

# Chemical composition of secondary organic aerosol particles formed from mixtures of anthropogenic and biogenic precursors

Yunqi Shao[1], Aristeidis Voliotis[1], Mao Du[1], Yu Wang[1], Kelly Pereira[3,*], Jacqueline Hamilton[3], M. Rami Alfarra[1,2 ‡], Gordon McFiggans[1]

[1]School of Earth and Environmental Science, University of Manchester, Manchester, M13, 9PL, UK
[2]National Centre for Atmospheric Science
[3] Wolfson Atmospheric Chemistry Laboratories, Department of Chemistry, University of York, York, YO105DD, UK
*Now at: Department of Life and Environmental Sciences, Bournemouth University, Dorset, BH12 5BB, UK
10   ‡ now at Environment & Sustainability Center, Qatar Environment & Energy Research Institute, Doha, Qatar

Correspondence to: Yunqi.Shao (Yunqi.Shao@Manchester.ac.uk)





**Abstract.**

A series of experiments were designed and conducted in the Manchester Aerosol Chamber (MAC) to study the photooxidation of single and mixed biogenic (isoprene and α-pinene) and anthropogenic (*o*-cresol) precursors in the presence of NOx and ammonium sulphate seed particles. Several online techniques (HR-TOF-AMS, Semi-Continuous GC-MS, NOx and $O_3$ analyser) were coupled to the MAC to monitor the gas and particle mass concentrations. Secondary Organic Aerosol (SOA) particles were collected onto a quartz fibre filter at the end of each experiment and analysed using liquid chromatography ultra-high resolution mass spectrometry (LC-Orbitrap MS). The SOA particle chemical composition in single and mixed precursor systems was investigated using non-targeted accurate mass analysis of measurements in both negative and positive ionization modes, significantly reducing data complexity and analysis time, providing an more complete assessment of the chemical composition. This non-targeted analysis is not widely used in environmental science and never previously in atmospheric simulation chamber studies. Products from α-pinene were found to dominate the binary mixed α-pinene / isoprene system in terms of signal contributed and the number of particle components detected. Isoprene photooxidation was found to generate negligible SOA particle mass under the investigated experimental conditions and isoprene-derived products made a negligible contribution to particle composition in the α-pinene / isoprene system. No compounds uniquely found in this system contributed sufficiently to be reliably considered as a tracer compound for the mixture. Methyl-nitrocatechol isomers ($C_7H_7NO_4$) and methyl-nitrophenol ($C_7H_7NO_3$) from *o*-cresol oxidation made dominant contributions to the SOA particle composition in both the *o*-cresol / isoprene and *o*-cresol / α-pinene binary systems in negative ionization mode. In contrast, interactions in the oxidation mechanisms led to the formation of compounds uniquely found in the mixed *o*-cresol containing binary systems in positive ionization mode. $C_9H_{11}NO$ and $C_8H_8O_{10}$ made large signal contributions in the *o*-cresol / isoprene binary system. The SOA molecular composition in the *o*-cresol / α-pinene system in positive ionization mode is mainly driven by the large molecular weight compounds (e.g. $C_{20}H_{31}NO_4$, and $C_{20}H_{30}O_3$) uniquely found in the mixture. The SOA particle chemical composition formed in the ternary system is more complex. The molecular composition and signal abundance are both markedly similar to those in the single α-pinene system in positive ionization mode, with major contributions from *o*-cresol products in negative ionization mode.

## 1. Introduction

### 1.1. Organic Aerosols and its impacts

Atmospheric aerosols affect climate directly through scattering or absorbing solar-radiation (Novakov and Penner, 1993; Andreae and Crutzen, 1997) and indirectly by acting as cloud condensation nuclei (CCN) (Mcfiggans et al., 2006). Exposure to particulate matter has also been directly linked to adverse impacts on human health (WHO, 2016). Organic aerosol significantly contributes to fine particulate matter (PM) in the atmosphere (Fiore et al., 2012; Jimenez et al., 2009), and can affect human health through the deep penetration of small aerosol particles into the lungs through inhalation, and the deposition of larger particles in the upper respiratory tract (Burnett et al., 2014). Fine PM has a wide variety of primary (e.g. agricultural





operations, industrial processes, and combustion processes) and secondary sources. In addition to secondary inorganic contributions from nitrate and sulphate, secondary organic aerosol (SOA) formed from the oxidation of atmospheric volatile

organic vapours (VOCs) can make a major contribution (Hallquist et al., 2009).

## 1.2. SOA and its formation pathways

The chemical diversity of volatile organic compounds (VOCs) and their oxidation pathways substantially influence SOA chemical composition  (Lim and Ziemann, 2009).  VOCs can be both anthropogenic and biogenic in origin (Li et al., 2018). Common and abundant anthropogenic VOCs include aromatic hydrocarbons such as benzene, toluene and cresol, emitted from

a wide variety of human activities, e.g.  cooking and biomass-burning (Atkinson and Arey, 2003), with the latter being an oxidation product of the former two compounds (Schwantes et al., 2017).  Biogenic VOCs, including isoprene and monoterpenes (e.g. α-pinene) are emitted in large quantities by vegetation, oceanic macroalgae and microalgae (Bravo-Linares et al., 2010; Atkinson and Arey, 2003).  Once emitted into the atmosphere, VOCs undergo oxidation by the prevailing atmospheric oxidants; the hydroxyl radical (OH) during daytime, the nitrate radical ($NO_3$) at night-time, and the unsaturated

fraction by ozone during both day and night (Atkinson, 1997). The oxidation of VOCs can result in the formation of both more and less volatile organic products (Jimenez et al., 2009). Low volatility organic products can condense onto existing particles or form new particles through nucleation if sufficiently low in volatility, as described by the gas-particle partitioning framework (Schervish and Donahue, 2020; Donahue et al., 2011).  VOC oxidation can result in a range of multi-functional products. Multiple generations of gas phase oxidation results in continually evolving chemical speciation either in the gas or

particulate phase (Mcneill, 2015; Shrivastava et al., 2017) and owing to the complexity of gaseous and particulate phase oxidation pathways, SOA formation mechanisms remain unclear and require further investigation .

## 1.3. Prior studies of using offline techniques

Whilst techniques for online or semi-continuous SOA compositional measurements have recently become more widely adopted (Zhang et al., 2011; Ahlberg et al., 2017; Schwantes et al., 2017; Hamilton et al., 2021; Lopez-Hilfiker et al., 2014;

Decarlo et al., 2006)), offline techniques generally provide more detailed insight into molecular composition. Offline techniques such as gas chromatography mass spectrometry (GC-MS) (Ono-Ogasawara et al., 2008; Saldarriaga-Noreña et al., 2018; Cropper et al., 2018), and liquid chromatography mass spectrometry (LC-MS) (Coscollà et al., 2008; Buiarelli et al., 2017; Pereira et al., 2015) can identify the chemical composition for thousands of organic compounds, with some of the techniques revealing information about compound's structure, alluding to  potential sources and formation mechanisms (Liu

et al., 2007; Singh et al., 2011; Ono-Ogasawara et al., 2008; Carlton et al., 2009; Kroll et al., 2005a; Ng et al., 2008; Nestorowicz et al., 2018; Eddingsaas et al., 2012). LC-MS has been widely employed for the chemical  characterisation of laboratory generated SOA and ambient SOA.  For example, targeted analysis of SOA products using high-performance liquid chromatography time-of-flight mass spectrometry  (HPLC-ToF-MS) illustrated a new pathway for the formation of 3-methyl-1,2,3-butane-tricarboxylic acid (MBTCA) through the further oxidation of nopinone, a known product in the oxidation of β-





pinene by OH (Mutzel et al. (2016). Hamilton et al. (2021) used targeted LC-Orbitrap MS analysis of ambient Beijing filter samples to identify tracers of isoprene nitrate formation pathways in both gas and particle phases indicating a strong dependence on nitrate radicals from early afternoon onwards. These targeted approaches are somewhat limited by their inability to comprehensively account for the entire mass of SOA components, though it is impractical to extract the non-targeted chemical information by manual data processing in complex ambient systems.  Non-targeted screening tools have been widely

employed in metabolite and protein analysis to reduce data analysis time but are uncommon in environmental science applications. Chromatographic separation coupled with Fourier transform mass spectrometers (e.g., Orbitrap) have sufficient mass resolution to characterise the chemical composition of complex particulate matter with the ability to distinguish structural isomers. Exploiting this capability, a methodology for automated non-targeted screening was presented by Pereira et al (2021) using ultrahigh-performance liquid chromatography–Orbitrap MS data. This non-targeted screening tool has been rigorously

tested using authentic standards, and provides molecular formula assignments and plausible structure information (among other information) for all detected compounds within a sample dataset. Moreover, the accurate mass spectrometry employed has a mass resolution of 70,000 at $m/z$ 200, leading to substantial increase in the signal/noise ratio and enhanced quantification of low concentration species. One of the few studies applying automated non-targeted method in environmental matrices, Mehra et al. (2021) used this approach for LC-Oribitrap MS data to characterise the SOA from the low-NOx oxidation of 1-

methylnaphthalene, propylbenzene and 1,3,5-trimethylbenzene in laboratory measurement, alongside characterise the SOA from filter that collected in urban area, which aims to to study the anthropogenic and biogenic contributions to organic aerosol. This study also compared the result with online technique a time-of-fight chemical ionisation mass spectrometer using an iodide ionisation system (I-CIMS), which show good agreement between observation of online I-CIMS results and results of offline LC-Orbitrap MS in negative ionization mode. Wang et al. (2021) also used non-targeted method for LC-Oribitrap MS

data to characterise particulate products on filters that collected from three cities located in northeast, east and southeast China, namely Changchun, Shanghai and Guangzhou. This study suggested that anthpogenic emissions are the dominate source of urban organic aerosol in all three cities. Also, they found out that samples from Shanghai and Guangzhou shared considerate chemical similarity, but significant differ from Changchun. In our present study, for the first time, we will apply this automated non-targeted screening tool for the compositional analysis of SOA generated in an aerosol chamber from single and mixed

precursor experiments.

### 1.4. Summary of studies on similar SOA systems

There are numerous studies investigating SOA formation from the oxidation of biogenic VOCs, particularly for terpenoid

compounds (Stroud et al., 2001; Surratt et al., 2006; Dommen et al., 2006; Carlton et al., 2009; Camredon et al., 2010; Surratt et al., 2010; Henry et al., 2012; Ahlberg et al., 2017; Hoffmann et al., 1997; Odum et al., 1996). Isoprene ($C_5H_8$) is the most abundant biogenic VOC emission and α-pinene ($C_{10}H_{16}$) is one of the most abundant and widely studied biogenic monoterpene



(Hallquist et al., 2009). Whilst oxidation products from these two biogenic precursors are both considered to contribute substantially to the global SOA budget, there are marked differences in their SOA particle mass yield; α-pinene has a yield in the range of 17 to 45% (Mcvay et al., 2016; Ng et al., 2007; Eddingsaas et al., 2012), while isoprene has a much lower yield in the range of 0 to5% (Dommen et al., 2006; Kroll et al., 2005a; Kroll et al., 2006; Pandis et al., 1991; Carlton et al., 2009). The reason for the low isoprene SOA yield is in part a result of the high volatility of oxidation products. However, the yield of isoprene SOA is strongly acid-dependent and closely related to the particle-phase acidity due to the impact on the amount of heterogenous uptake, which might be the reason found higher isoprene SOA mass concentration when increasing aerosol acidity.(Surratt et al., 2007a) . Xu et al. (2021) demonstrated that over 98% of isoprene oxidized organic molecules by mole were classified as semi-VOC (SVOC) and intermediate-VOC (IVOC) with volatility ($\log_{10}C^*$, ug m$^{-3}$) range of -0.5 to 5, while about 1.3% of isoprene oxidation products were considered as low-VOC (LVOC). Conversely, the larger $C_{10}$ monoterpene skeleton of α-pinene typically results in the formation of less volatile oxidation products. Lee et al. (2021) reported that the SOA from α-pinene ozonolysis required 80 ºC for complete volatilisation, and the volatility of α-pinene SOA strongly depended on the VOC/NOx ratios, forming volatile nitrate containing species under high NOx conditions.

There are many studies reporting the chemical characterisation of SOA formed in smog chambers from α-pinene and isoprene using liquid chromatography mass spectrometry (LC-MS). (Yasmeen et al., 2012; Surratt et al., 2006; Kahnt et al., 2014; Pereira et al., 2014; Winterhalter et al., 2003). Winterhalter et al. (2003) used LC-MS to demonstrated thet major particulate phase compounds from the O₃ and OH oxidation of α-pinene, such as cis-pinic acid, cis-pinonic acid, hydroxy-pinonic acid isomers, and possibly hydroxy-carboxylic acid. It is worth noting that this study suggested the ozonolysis reaction is the main driven pathway of aerosol formation regard to its performance of various experiments. Similarly, Surratt et al. (2006) studied isoprene photooxidation under various NOx conditions. The chemical composition of isoprene SOA products was analysed by a series of online and offline techniques (including LC-MS) and indicated that oligomerisation plays an important role in SOA formation pathways, especially under high NOx conditions, forming acidic products.

SOA can also be produced from anthropogenic VOCs (e.g. o-Cresol), although global biogenic SOA production (~88 TgC per year) is thought to dominate over the anthropogenic SOA production (~10 TgC per year) (Hallquist et al., 2009). Schwantes et al. (2017) studied the formation of low-volatility products from *o*-cresol photooxidation under various NOx conditions using chamber experiments with chemical ionization mass spectrometry (CIMS) and direct analysis in real time mass spectrometry (DART-MS). This study identified several *o*-cresol oxidation products, including the first generation product (methyl-catechol), second generation products (trihydroxy-toluene and hydroxy-methyl-benzoquinone) and third generation products (tetrahydroxy-toluene and dihydroxy-methyl-benzoquinone), indicating successive addition of OH radicals onto the aromatic ring during the oxidation, following expected mechanistic pathways (Atkinson and Aschmann, 1994; Olariu et al., 2002)





Despite the wealth of knowledge of gaseous and particulate phase product formation from the oxidation of single VOC precursors using chamber experiments, there is a comparative lack of understanding in the real atmosphere. Online measurements of the OA composition by Aerodyne High Resolution Aerosol Mass Spectrometer (HR-TOF-AMS) and VOC by Ionicon Proton Transfer Reaction Mass Spectrometer (PTR-MS) during the CARES campaign in the vicinity of

Sacramento, California indicated that the mixing of anthropogenic emissions from Sacramento with isoprene-rich air from the foothills enhance the production of OA (Shilling et al., 2013). This study suggested anthropogenic/biogenic interactions enhances OA production from biogenic species, suggesting the amount of isoprene SOA strongly depends on VOC/NOx ratio. However, the physical and chemical reasons for such interactions remain unclear and warrant further investigation. There have been several laboratory studies investigating the SOA formation in mixed VOC systems. Ahlberg et al. (2017) investigated

SOA from VOC mixtures including biogenic (α-pinene, myrcene and isoprene) and anthropogenic VOCs (m-xylene) in an oxidation flow reactor (OFR) equipped with high-resolution time-of-flight aerosol mass spectrometry (HR-ToF-AMS). Their results showed that the SOA mass yield formed from a VOC mixture containing myrcene was higher than expected, possibly a result of myrcene nucleating particles leading to an increased condensation sink under the conditions of the OFR. This study also found that the SOA particle size was larger in VOC mixtures with isoprene and unlimited oxidant supply. However, other

studies indicate that isoprene could inhibit new particle formation by scavenging oxidant and forming relatively high volatility organic products than nucleating precursors (Kiendler-Scharr et al., 2009; Kiendler-Scharr et al., 2012). Mcfiggans et al. (2019) reported a reduction in SOA mass and yield from the VOC mixture of α-pinene and isoprene with an increasing fraction of isoprene in the mixture. This was attributed to isoprene acting as an OH scavenger and its radical oxidation products reacting with those formed from α-pinene, enhancing the overall volatility of the products in the mixture. This study indicates that

interactions between VOC products should be considered to enable a mechanistic understanding of SOA formation in the ambient atmosphere. Shilling et al. (2019) reported that freshly formed isoprene-SOA did not fully mix with pre-existing SOA in isoprene/ α-pinene mixture system (e.g. aged isoprene-SOA and aged α-pinene SOA) over the 4 hours experimental time scale in sequential condensation experiment, without observing notable suppression of SOA formation in α-pinene/isoprene mixture system.


### 1.5. This present study

In this study, we designed a series of chamber experiments using single, binary and ternary VOC systems, expanding on the work performed by McFiggans et al., (2019), with the aim of better understanding the chemical composition and interactions during SOA formation in mixed VOC systems. We move beyond the consideration of SOA formation from anthropogenic

VOC precursors to consider the effect of their mixture with biogenic VOC. Ortho-cresol (o-cresol) was chosen as an anthropogenic precursor with a moderate SOA yield, with a comparable reactivity towards the hydroxyl radical (OH) and a negligible reactivity towards ozone. We retained the two biogenic precursors studied in McFiggans et al. (2019); isoprene





being the dominant VOC emitted from plants globally, but with modest SOA formation potential and alpha-pinene ($\alpha$ -pinene), similarly widely emitted in a lower amounts, but a more efficient SOA precursor.


The objectives of present study are to investigate, using offline analysis of SOA chemical composition, whether (i) high yield precursors dominate the contribution to SOA formation of mixture systems and (ii) cross-products from mechanistic interactions in the oxidation of precursors feature strongly in the mixed precursor systems. A series of photochemical oxidation experiments were designed and conducted to produce SOA from the selected VOCs (α-pinene, isoprene and $o$-cresol) and their

mixtures in the presence of neutral seed particles (ammonium sulphate) and NOx. The experimental programme included three single precursor systems, three binary precursor mixtures and one ternary mixture of precursors. The aerosol samples were collected onto a filter from each experiment and analysed offline using liquid chromatography ultra high resolution mass spectrometry with an automated non-targeted data processing methodology recently described in Pereira et. al (2021).

## 2. Method

### 2.1. Chamber description

All experiments were performed in the 18m$^3$ Manchester Aerosol Chamber (MAC). Briefly, the MAC operate as a batch reactor to study the atmospheric processing of multicomponent aerosols under controlled conditions. The chamber comprises an FEP Teflon bag mounted on three rectangular extruded aluminium frames, housed in an air-conditioned enclosure. Two 6

kW Xenon arc lamps (XBO 6000 W/HSLA OFR, Osram) and a bank of halogen lamps (Solux 50 W/4700 K, Solux MR16, USA) are mounted in the inner aluminium wall of the enclosure which is lined with reflective "space blanket" material to provide maximum and homogenous light intensity to simulate the realistic day-time atmospheric environment. To remove unwanted radiation flux below 300 nm, a quartz filter was mounted in front of each arc lamp. Removal of unwanted heat from the lamps and temperature and relative humidity control of the chamber was assisted by conditioned air introduced between

the bag and the enclosure at 3 m$^3$ s$^{-1}$ and active water cooling of the mounting bars of the halogen lamps and of the filter in front of the arc lamps. Regular steady state actinometry experiments were conducted through the entire campaign and indicated that the photolysis rate of NO$_2$ ($J$NO$_2$) in a range of 1.83-3 x10$^{-3}$s$^{-1}$ during experimental period. Photolysis of NO$_2$ leads to O$_3$ formation, which further photolyses to produce OH radicals in our moist experiments. Humidity and temperature are controlled by the humidifier and by controlling the air conditioning set-point during the experiment and continuously monitored using a

dewpoint hygrometer and a series of thermocouples and resistance probes throughout the chamber. Additional online instruments included a semi-continuous gas chromatography mass spectrometer (GCMS) for VOC measurement (Minaeian, 2017), a water-based condensation particles counter, a differential mobility particle sizer (DMPS) and an aerosol mass spectrometer (AMS) for particulate-phase compound measurement (Canagaratna et al., 2007). The filter collection, extraction,



measurement and analysis techniques are described below. Full details of the MAC characterisation, the experimental
procedure and instrumentation payload is provided in Shao et al. (2022).

### 2.2. Experimental Strategy

The experimental programme was conceived using a concept of "initial iso-reactivity" towards OH, with the intention of
allowing a reasonably comparable contribution of oxidation products from each VOC at the chosen concentration and
experimental conditions. Clearly this does not take into account consumption by oxidants other than OH formed during the
experiment (notably ozone) and also neglects the reactivity of the subsequent oxidation products. The injected precursor mass
was therefore chosen according to its reactivity towards OH (Atkinson, 2004). SOA composition was determined using
analysis of chamber filter samples by liquid chromatography ultra-high resolution mass spectrometry (LC-Orbitrap MS) and
automated non-targeted data processing for all single precursor and mixed VOC systems.


### 2.3. Experimental Procedure

Programmed "pre-experiment" and "post-experiment" procedures were routinely conducted before and after each SOA
experiment to minimize the possible contamination in the chamber. The "pre-experiment" and "post-experiment" are
comprised of multiple automated fill/flush cycles with an approximate air flow rate of 3m$^3$ min$^{-1}$, for cleaning the chamber.
The upper and lower frames were free to move vertically to expand and collapse the bag during the fill/flush cycle. Filtered
air was sequentially injected into and extracted from the bag, reducing contaminants in the bag with each cycle. Several
instruments (e.g. WCPC, Model 49C O$_3$ analyser, Thermo Electron Corporation) and Model 42i NO-NO$_2$-NOx analyser,
Thermo Scientific) were continuously connected to the chamber during the pre-experiment to monitor the concentration of
particles, concentration of ozone and NOx, and to ensure the bag was sufficiently clean (with all aforementioned factors close
to zero) to conduct the chamber background procedure. When conducting this procedure, there were no reactants in the bag
and the bag was stabilised for at least an hour for the instruments to establish the baseline of the clean chamber. In the following
stage, VOC precursor(s), NOx and seed particles were injected into the chamber sequentially in dark conditions and the
chamber remained steady for an hour for the instruments to obtain a baseline of the initial chamber conditions (e.g.,
experimental background) before the SOA experiment. The baselines of chamber background and experimental background
were subsequently subtracted from the experimental measurements.

Ammonium sulphate seed particles were generated via atomization from ammonium sulphate solution (Puratonic, 99.999%
purity) using a Topaz model ATM 230 aerosol generator. The concentration of seed particles in the chamber was controlled
by altering the injection time and concentration of the prepared solution (0.01g/ml). The accumulating seed particles injected





into the stainless-steel residence chamber for 1 min then diverting the main chamber injection flow for 30s during the final fill cycle of pre-experiment procedure. The liquid α-pinene, isoprene and *o*-cresol (Sigma Aldrich, GC grade ≥99.99% purity) were injected as required through the septum of a heated glass bulb and evaporated into an $N_2$ carrier flow into the chamber during this final fill along with $NO_x$ as $NO_2$ from a cylinder, also carried by $N_2$. The injected VOC mass was calculated using the "initial OH isoreactivity" approach described above. Photochemistry was initiated by irradiating the VOC at a moderate

VOC / $NO_x$ ratio using the lamps as described above. The concentration of $NO_x$ and $O_3$, particles number concentration and mass concentration were monitored during the experiment using the online instruments. SOA particles were collected on a blank filter (Whatman Quartz microfiber, 47 mm) mounted in a bespoke holder built into the flush pipework by flushing the remaining chamber contents after a 6-hour experiment. The filters were then wrapped in foil and stored at -18°C prior to analysis. Quartz fibre filters were pre-conditioned by heating in a furnace at 550°C for 5.5 hours. Actinometry and off-gassing

experiments were conducted regularly after several of SOA experiments to establish the consistency of the chamber's performance, evaluate the effectiveness of the cleaning procedure and confirm cleanliness of the chamber. "Background" filters were collected from the actinometry and off-gassing experiments. A summary of experimental conditions is given in Table 1.

**Table 1: Experimental descriptions, VOC mixing ratios, VOC:NOx ratio and mass concentration of seed particles in chamber.**

| Experiment type | Experiment | Experimental conditions | | |
| --- | --- | --- | --- | --- |
| | | Nominal VOC (ppbv) | Nominal VOC:NOx | Mass conc. $(NH_4)_2SO_4$ (ug/m$^3$) |
| **Single Precursor** | (a) | α-pinene: 309 | 7.7 | 72.6 |
| | (b) | Isoprene: 164 | 7.1 | 101.9 |
| | (c) | *o*-cresol: 400 | 9.1 | 47.8 |
| **Mixed Precursors (Binary)** | (d) | α-pinene: 155 Isoprene: 82 | 9.9 | 50.5 |
| | (e) | α-pinene: 155 *o*-cresol: 200 | - | 42.5 |
| | (f) | Isoprene: 82 *o*-cresol: 200 | 8.3 | 49.6 |
| **Mixed Precursors (Ternary)** | (g) | α-pinene: 103 Isoprene: 55 *o*-cresol: 133 | 3.7 | 45.8 |



### 2.4. Offline Analysis of the Filter Samples

#### 2.4.1.    Sample preparation

Filter samples of SOA particles were extracted using the following procedure.  Each filter was cut into small pieces into pre-
cleaned 20 mL scintillation vial. 4 mL of (Fisher Scientific FB15051of methanol  (Optima LC-MS grade, ThermoFisher
Scientific) was added to the vial. The sample was then wrapped in foil and left for 2 hours at ambient temperature, sonicated
for 30 minutes and the extractant  filtered through a 0.22 µm pore size PDVF filter using a BD PlasticPak syringe.  An
additional1 ml of methanol was added to the vial and filtered through the same syringe membrane to minimise sample loss. .
The filtered extractant was then evaporated to dryness using solvent evaporator (Biotage, model V10) at 36°C and 8 mbar
pressure and redissolved in 1 ml of a 90:10 water: methanol (Optima LC-MS grade)  for LC-Orbitrap MS analysis.

#### 2.4.2.    Liquid Chromatography Mass Spectrometry Analysis

Samples were analysed using ultra-performance liquid chromatography ultra-high resolution mass spectrometry (Dionex 3000,
Orbitrap QExactive, ThermoFisher Scientific). A reverse-phase C18 column (aQ Accucore, ThermoFisher Scientific) 100 mm
(long) × 2.1 mm (wide), with a 2.6 µm particle size was used for compound separation. The flow rate was set to 0.3 ml/min,
with 2µL sample injection volume. The   autosampler temperature was set to   4ºC and the column at 40ºC. The mobile phase
solvent included (A) water and (B) methanol that both contain 0.1% (v/v) formic acid (Sigma Aldrich, 99% purity).  Gradient
elution was performed starting at 90 % (A) with a 1-minute post-injection hold, decreasing to 10% (A) over 26 minutes, before
returning to the initial mobile phase conditions at 28 minutes, followed by a 2 minute column re-equilibration.  Electrospray
ionisation (ESI) was used with a mass-to-charge ($m/z$) scan range of 85 to 750. The ESI parameters were set as follows: 320ºC
for capillary and auxiliary gas temperature , 70 (arbitrary units) and 3 (arbitrary units) flow rate for sheath gas and auxiliary
gas respectively. (Pereira et al., 2021).  Compound fragmentation was achieved  using higher-energy collision induced
dissociation ($MS^2$). A fragmentation spectrum is generated for each selected precursor, which  allows structural identification
through the elucidation of fragmentation patterns (Mcluckey and Wells, 2001). This fragmentation spectra can aid in the
structural identification of  isomeric species (i.e. compounds with the same molecular formula, but different structural
arrangement).  Accurate mass calibration was performed prior to sample analysis in positive and negative ESI mode using the
manufacturer recommended  calibrants ( Thermo Scientific). A procedural control  (i.e. pre-conditioned blank filter subject to
the same sample extraction procedure) was analysed, along with solvent blanks (consisting of 90:10 water: methanol) which
were frequently run throughout the sample analysis sequence, allowing any instrument or extraction artefacts to be detected.
Automated non-targeted data analysis was performed using Compound Discoverer version 2.1 (Thermo Fisher Scientific). Full
details of the data processing methodology can be found in Pereira et al. (2021). Briefly, the chemical information of all
detected compounds in each sample data file are extracted.  The method provides molecular formulae assignment of detected
compounds using the following elemental restrictions:  unlimited carbon, hydrogen and oxygen atoms, up to 5 nitrogen and





sulphur atoms, and in positive ionisation mode, 2 sodium and 1 potassium atom are also allowed. Molecular formulae were
attributed if the mass error < 3 ppm, signal-to-noise ratio > 3, and the isotopic intensity tolerance was within ± 30 % of the
measured and theoretical isotopic abundance. Instrument artefacts and compounds detected in the "background" filter were
removed from sample data if the same detected molecular species had a retention time within 0.1 minutes and sample/artefact
or background peak area ratio < 3.  Any compounds detected in the sample and background data with a sample/background
peak area ratio >3 were conserved in sample data set after subtracting the background peak area (new peak area = sample peak
area – background peak area). The automated Python program generates a list of detected compounds, assigned molecular
formulae and  tentatively assigned mass spectral library identifications (see Pereira et. al (2021) for further information).  The
mass spectra for both ESI  modes from each VOCs system, are shown in supporting information (Figure S1 and S2).

To provide confidence in the components in each system detected by the non-targeted method,  only those compounds found
in all replicate experiments and not found in any background "clean" experiments were attributed to a particular single
precursor or mixed system. The approach taken thus ensures the most conservative assignment of compounds to a particular
precursor system. Where quantities are analysed and presented from "representative" experiments, only those relating to
compounds found in all replicate experiments are confidently attributed to this particular system. Compounds that were found
above detection limit in only a subset of the experiments in a single system were not attributed to the system and were
considered "inconclusive". Moreover, the common compounds were only considered to be the same detected molecular species
if they had a retention time within 0.1 minutes and sample/artefact peak area ratio > 3 in all replicate experiments. Section
3.2.1 and section 3.2.3 only considers the compounds which can be confidently attributed to a particular system. For the
elemental characterization in section 3.2.2, both the confident and inconclusive components are presented, with only the
compounds confidently attributed analysed according to carbon number.

## 3.  Results and Discussion

### 3.1. SOA particle mass formation in the experiments

The formation of SOA particle mass in the seven experimental systems is shown alongside the VOC concentration, NOx and
$O_3$ mixing ratio time series in Figure 1. As shown in figure 1(a), the particle wall-loss corrected SOA mass in all α-pinene
containing systems reaches a maximum value within the 6-hour experimental timeframe. α-pinene produced the highest SOA
particle mass (~400ug/m$^3$) of all systems at nominal "full" VOC reactivity with the most rapid onset and rate of mass formation.
The SOA particle mass continued to increase at the end of the experiment in the single VOC *o*-cresol and binary isoprene/*o*-
cresol systems. No measurable SOA particle mass above background (~0 ug/m$^3$) was produced within the 6-hour duration in
any single VOC precursor isoprene experiment.





As shown in Fig.1 (b), NOx was observed to decay in all systems where significant SOA mass was formed, but little NOx consumption was observed in the single isoprene system or in the binary isoprene / o-cresol mixture. The reduction of NOx will result from i) reaction between OH radicals and $NO_2$ leading to $HNO_3$ formation with subsequent loss to the chember walls or particles as inorganic nitrates. ii) termination reactions between NO and $RO_2$ radicals or $NO_2$ and $RO_2$ radicals leading to formation of nitrogen-containing organic ($NOROO_2$ and $NO_2ROO_2$) compounds(Atkinson, 2000).


Noting that there was no $O_3$ initially in any experiment, Fig. 1(c) illustrates ozone concentration time series in each system. Ozone can be seen to increase during the initial stage of experiment in most experiments, with most modest rises in the single o-cresol and binary isoprene/o-cresol systems. An initial rise is expected owing to the fairly rapid photolysis of $NO_2$ tending towards photo-stationary state (PSS) between $NO_2$, NO and $O_3$. The onset of VOC oxidation will result in a consumption of

$O_3$ when the unsaturated α-pinene and isoprene are present. At the same time NO will react with $RO_2$ and $HO_2$ radicals formed in the VOC degradation, resulting in $NO_2$ and OH radical formation. The reduction in the proportion of NO reacting with $O_3$ and photolysis of the $NO_2$ produced results in net $O_3$ production and deviation from PSS.

The time profile of the VOC concentration from experiments in all single and mixed precursor systems are shown in Fig.1 (e-

f). A rapid and pronounced onset of VOC consumption in each system is observed after illumination, attributable to reaction with OH radicals, and $O_3$ in α-pinene and isoprene-containing systems. Panels (d) to (f), plotted logarithmically for clarity, show the VOC decay profile in each experiment reflecting their differences in reactivity and the variable oxidant regime in each experiment. Individual VOCs have comparable decay rates in each mixture except for i) α-pinene in the binary α-pinene / o-cresol system, which had a significantly lower decay rate than it had in other α-pinene-containing systems systems and ii)

isoprene, which had a faster decay rate in the binary α-pinene / isoprene system than in other isoprene-containing systems. No VOC was entirely consumed in any system by the end of the 6 hours experiments, with consumption continuing until the end.



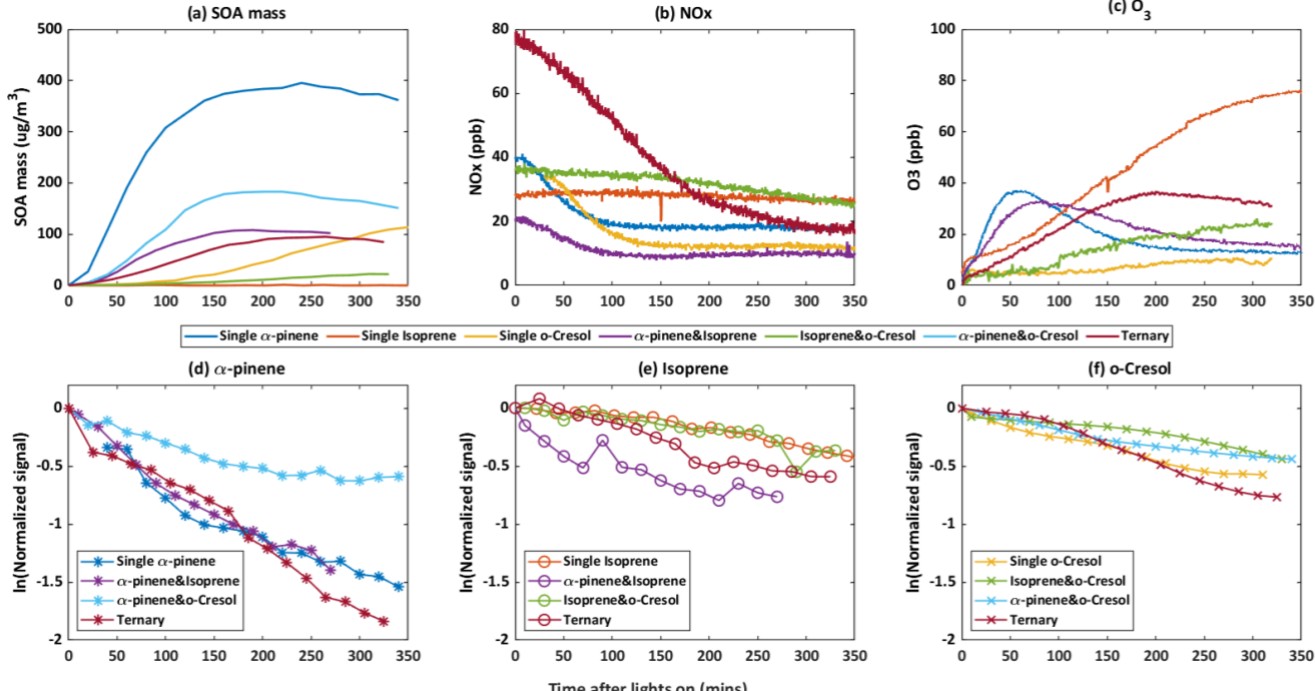

**Figure 1:Evolution of gas and total SOA particle mass measurements during the photo-oxidation of VOCs after chamber illumination. (a) The SOA mass was measured using a high-resolution time-of-flight aerosol mass spectrometer (HR-ToF-AMS) during single, binary and ternary experiment. (b)–(c): Concentration of NOx and O3 against time in all of single, binary and ternary experiments (data was unavailable for the binary α-pinene/o-cresol experiment). (d)-(f): decay rata of VOC across all systems in. α-pinene (b), isoprene(c) and o-cresol(d) in single, binary and ternary experiments respectively.**

## 3.2. Characterisation of Components by LC-Orbitrap MS

### 3.2.1.    Characterisation by number of discrete compounds in each system

The number of discrete peaks extracted using the Compound Discoverer software from the LC-Orbitrap MS data for all experiments in each SOA system is listed in Table 2 and illustrated using Venn diagrams showing the compounds found in more than one system (henceforth referred to as "common" compounds) and those found solely in a single system (referred to as "unique") in Figures 2 and 3 (in negative and positive ionisation modes respectively).

As seen in Table 2, all α-pinene-containing systems were found to contain a greater number of compounds than any system not containing α-pinene. The binary α-pinene / isoprene system contained the highest number of all systems, with 377 in negative ionization mode and 441 in positive ionization mode. A total of 644 total compounds were seen in the single VOC α-pinene system across both negative and positive ionization modes, fewer than in the binary α-pinene/isoprene system with 818





compounds, but higher than the α-pinene/ *o*-cresol  system with 483 compounds. The total number of discrete products in the ternary system is lower than in the single α-pinene and binary α-pinene system.  The single VOC isoprene system generated the lowest total number of products of all systems above detection limit. This is unsurpriseing, since undetectable mass concentration was found by the online instrumentation in these experiments. Multifunctional compounds can be detected in

both negative and positive ionization mode. Negative ionization mode typically exhibits high sensitivity towards compounds containing alcohol and carboxylic acid functionalities,  whereas positive ionization mode typically has a greater affinity for compounds with functional groups that are readily protonated (e.g. -NH, -O- or -S-, -CH2-,-C=O,-SO$_2$- group) (Glasius et al., 1999; Steckel and Schlosser, 2019).

**Table 2: Number of compounds detected in SOA sample in negative and positive ionization mode from single, binary and ternary precursor's system.**

| Experiment | Number of Detected Compounds | |
| :---: | :---: | :---: |
| | **Negative Mode** | **Positive Mode** |
| α-pinene | 282 | 362 |
| Isoprene | 28 | 68 |
| o-cresol | 84 | 53 |
| a-pinene/Isoprene | 377 | 441 |
| a-pinene/o-cresol | 339 | 144 |
| o-cresol/Isoprene | 72 | 87 |
| a-pinene/Isoprene/o-cresol | 112 | 188 |

**a) Negative Ionization Mode**

Fig. 2 shows a Venn diagram of the number of discrete compounds identified in negative ionisation mode in each of the individual and binary precursor experiments.  Figure. 2(a) and Figure 2(b) show that the number of discrete compounds from α-pinene dominated those found in the binary mixture system compared to those from the other precursors. 182 compounds found in all α-pinene single precursor experiments were also found in the binary α-pinene / isoprene mixed system; approximately 45 times greater than the 4 compounds also found in all single isoprene experiments. Similarly, 99 common

compounds were found between the single precursor α-pinene experiments and  those found in the binary α-pinene/ *o*-cresol system; roughly three times higher than the number of *o*-cresol-derived products that were also found in binary mixed system. More than half of the total number of compounds in the α-pinene / isoprene and α-pinene / *o*-cresol binary systems were unique to the mixtures and not observed in any of single precursor experiments. In the isoprene / *o*-cresol system a lower total number





of compounds were detected in every repeat experiment, with more compounds in the mixture also found in the *o*-cresol system
than the isoprene system (Fig.2(c)).

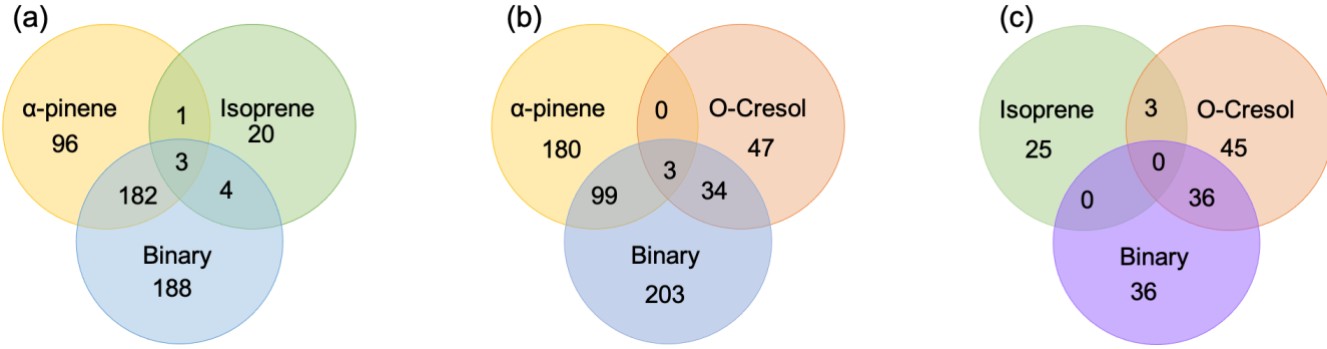

**Figure 2: Number of common compounds and unique compounds in single and binary precursors mixed experiments detected by**
**negative ionization mode LC-Orbitrap MS. Product are considered identical in the mixed and single precursor systems if the**
**compound has the same empirical formula and a retention time difference <0.1min.**


**b) Positive Ionization Mode**

Fig.3 shows the number of discrete SOA compounds identified in positive ionisation mode in the single and binary systems.
There are 226 compounds found in all α-pinene single precursor experiments that were also found in the binary α-pinene /
isoprene system; about 32 times more than also found in the isoprene-only experiments (Fig.3(a)). 48 α-pinene derived
compounds were also found in binary α-pinene/ *o*-cresol system; 16 times greater than those also found in all *o*-cresol only
experiments (Fig.3(b)). In both α-pinene containing binary mixtures, around or more than half of all detected compounds were
unique to the mixture. In the binary isoprene / *o*-cresol system shown in Fig.3(c)), *o*-cresol derived compounds were more
numerous than those in the isoprene experiments, with 23 compounds observed.






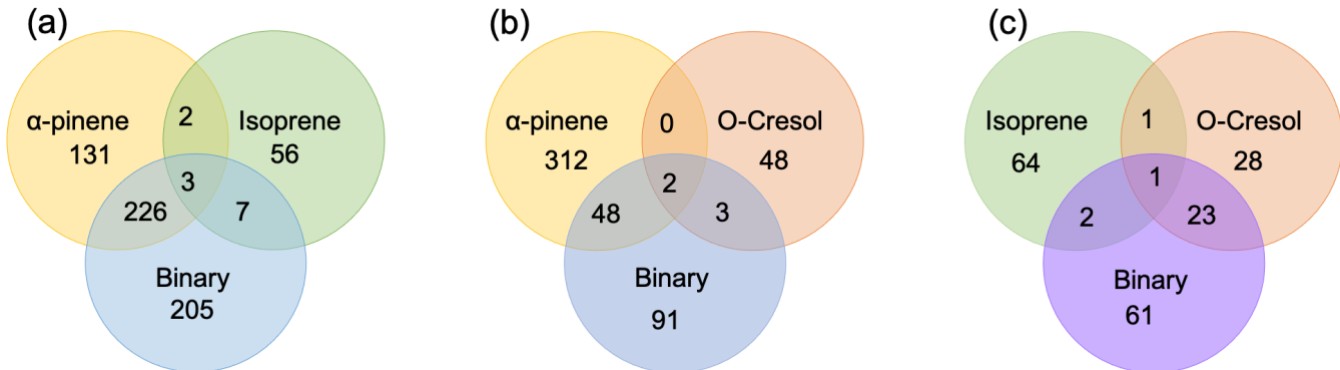

**Figure 3: Number of common compounds and unique compounds in single and binary precursors mixed experiments detected by positive ionization mode LC-Orbitrap MS. Products are considered identical in mixed and single precursor systems if a compound has the same empirical formula and the retention time difference <0.1min.**

The Venn diagrams for both ionization modes indicate the importance of α-pinene oxidation products in both binary systems, with large number of binary SOA compounds found to be present in the single precursor α-pinene system. In contrast, there are few common compounds observed between single isoprene and binary systems, possibly a result of the majority of isoprene derived products remaining in the gas-phase, or the isoprene products participating in cross-product formation in mixed precursor systems.

**3.2.2.    Characterization of Organic Particulates by Elemental Groups**

**3.2.2.1.  Negative Ionization Mode**

Elemental groupings are used here to provide insights into the SOA chemical composition in each system. All  detected molecular formulae in each system were classified into the following four categories based on their elemental compositions:

CHO, CHON, CHOS, and CHONS (C, H, O, N and S corresponding to the atoms in the molecule) and separated into seven carbon number categories.  The measured peak area of each compound was normalised to the total sample peak area as shown in Fig.4 and described in Pereira et. al (2021). Fig.4 presents the signal fraction of compounds in representative experiments that can be confidently attributed as found in each of the systems (i.e. that are found in every repeat experiment in this system) in the coloured stacked bars according to their carbon number and classified according to their elemental groupings. The

fractional contributions of compounds that confidently stated are similar for each experiment in a particular system. Also shown in the grey bar is the signal fraction of compounds that are inconclusively found in the experiment in each system classified by elemental grouping, but not found in all repeat experiments and chamber background experiment.





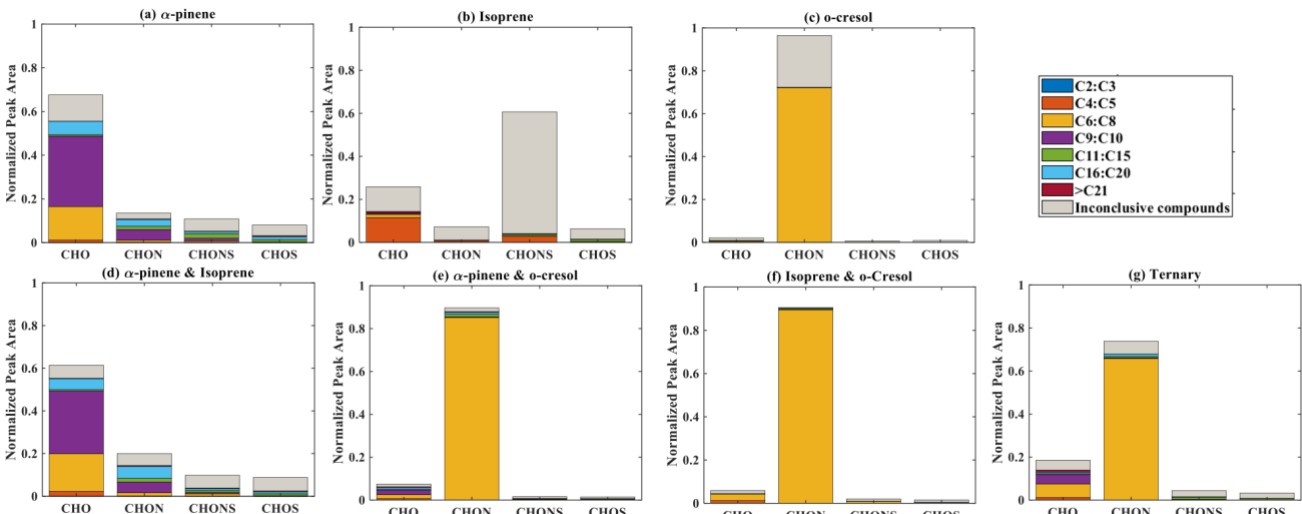

**Figure 4:The normalized signal intensity distribution of different compound categories (CHO, CHON, CHOS, and CHONS) for various single and mixed precursor systems in Negative Ionisation mode ESI (-) by LC-Orbitrap MS. The grey bar (inconclusive compounds) signal attributed to compounds that were not universally found in all repear experiments.**

**a) α-pinene**

As shown in in Fig 1(a), the SOA particle mass produced in the single precursor α-pinene system was greater than in any other system, at ~362 μg/m³. In the α-pinene single precursor representative experiment, ~ 55.6% of signal was found in molecules containing only C, H and O atoms with the majority consisting of 6 to 10 carbon atoms (47.5 %). Larger compounds were also observed with carbon numbers ranging  from $C_{16}$ and $C_{20}$ (representing 6.1 % of the total signal fraction) (Fig.4a)). Compounds

confidently found in ths system in the CHON , CHONS and CHOS groupings represented 10.8%, 5.2% and 3.2% of the signal abundance, respectively, again concentrated at $C_9$-$C_{10}$ and $C_{16}$-$C_{20}$. $C_{11}$-$C_{15}$ molecules represent 2.2% and 1.1% of the signal in the CHONS and CHOS categories respectively. Inconclusively attributed compounds contributed 25% of the total signal abundance; 47.6% of inconclusively compounds containing only C,H, and O atoms.

$C_6$ to $C_{10}$ compounds will include those produced through both functionalization (addition oxygenated function group) and fragmentation (cleavage of C-C bond) pathways during α-pinene oxidation (Eddingsaas et al., 2012). It has been further suggested that particle-phase dimerization and oligomerization reactions (e.g alcohol + carbonyl to form hemiacetals and acetals, hydroperoxide + carbonyl to form peroxyhemiacetals and peroxyacetals, carboxylic acid + alcohol to form esters, and aldehyde self-reactions to form aldols) can play an important role in α-pinene oxidation(Ziemann and Atkinson, 2012; Gao et

al., 2004a; Gao et al., 2004b), resulting in formation of large molecules (nC>10), potentially accounting for the $C_{16}$ to $C_{20}$



abundance. Recent studies have additionally identified gas-phase autoxidation as playing a pivotal role in formation of highly oxygenated molecules (HOMs)(Tomaz et al., 2021; Crounse et al., 2013; Bianchi et al., 2019; Zhao et al., 2018). HOM may condense on exiting seed particles or lead to new particle formation, depending on their vapour pressure (Tröstl et al., 2016). Autoxidation of $RO_2$ radicals in the gas-phase occurs rapidly via inter/intramolecular hydrogen abstraction leading to forming

R radicals with subsequent $O_2$ addition (Mentel et al., 2015; Jokinen et al., 2014). The new $RO_2$ radicals can undergo further autoxidation reaction, or react with $RO_2$ to generate dimer accretion products (Zhao et al., 2018; Berndt et al., 2018). Autoxidation may therefore contribute to CHO products with carbon numbers 16 – 20 in α-pinene oxidation. It has also been found that the uptake of α-pinene oxidation products on ammonium sulphate particles can lead to formation of organosulphate and nitrooxy organosulphate (Eddingsaas et al., 2012; Iinuma et al., 2009), contributing to the CHOS and CHONS groupings.


**b) isoprene**

As also seen in Fig.1(a), negligible SOA particle mass was generated in the single precursor isoprene system (~0 μg/m$^3$, close to our chamber background) and the total signal in Fig 4(b) therefore corresponds to extremely low SOA particle mass. Nevertheless, the presence of compounds in all repeat experiments but not on any filters taken in background experiments,

allows identification and attribution to isoprene products. Similar to the α-pinene system, compounds found in all repeat experiments containing CHO were the most abundant in the single precursor isoprene experiment shown in Figure 4(b), with normalized sample abundance of 14.3 %, mainly comprising compounds with 4 or 5 carbon atoms. Similarily, compounds in the CHONS classification can be confidently stated to make a non-negligible contribution to the total signal fraction, with a normalised abundance of 4.1%, also mainly comprising $C_4$-$C_5$ compounds. CHON (1%) and CHOS (1.5%) each contributed

significantly less than the other molecular groupings. Fig.4(b) shows that the majority of the signal in this single isoprene representative experiment was composed of compounds (78.9% nomalized of total signal fraction) that were not found in all isoprene experiments (and / or were also detected in background filters) and are therefore inconclusively assigned.

The presence of $C_4$-$C_5$ CHO compounds in single isoprene photo-oxidation system can be readily explained by established

oxidation pathways. For example, it is well-known that the double bond in isoprene is oxidized to form $C_4$ and $C_5$ compounds, such as methacrolein ($C_4$) and $C_5$-hydroxycarbonyls as first-generation products, and 2-methylglyceric acid ($C_4$) and isoprene tetrol ($C_5$) as second-generation products (Wennberg et al., 2018; Stroud et al., 2001; Carlton et al., 2009). However, it is less clear how such small compounds readily partition to the particle phase, owing to their relatively high vapour pressures, though it has been suggested that small compounds such as glyoxal (CHOCHO) have extremely high activity coefficients when

partitioning to aqueous particles, leading to low effective vapour pressures (Volkamer et al., 2009).

The negligible SOA particle mass formed in the isoprene single precursor system is consistent with the literature observations (Kroll et al., 2005a; Kroll et al., 2005b; Kroll et al., 2006; Carlton et al., 2009). However, condensed phase reactions on acidic seeds would be expected to appreciably increase this yield (Surratt et al., 2010; Surratt et al., 2007a; Carlton et al., 2009). The



large normalised signal contribution corresponds to the high number of inconclusively assignable compounds detected in this system. Most of these inconclusive compounds contained a large number of carbon atoms (nC>15).  These compounds are likely to have been formed via particle-phase accretion reactions, such as oligomerisation and organosulfate formation, even in the absence of acidity in our experiments, leading to low volatility higher molecular weight accretion products (Berndt et al., 2019; Carlton et al., 2009). Whether these products are formed on the filter medium or are present in the suspended particle

mass requires investigation. While these components are the most abundant, this still corresponds to a very small mass compared to all other systems and were not found in all repeat experiments.

### c) *o*-cresol

The particle wall-loss corrected SOA mass concentration at the end of the presented *o*-cresol experiment was approximately

101 µg/m$^3$ (Fig 1. (a)).  Fig.4(c) shows ~26.6% of the normalized signal abundance in inconclusively assigned compounds, mainly in the CHON classification. However, the key characteristic in the single precursor *o*-cresol system is that the most abundant compounds that are confidently found in all repeat experiments were found in the CHON category with between 6 and 8 carbon atoms (Fig 4(c)) with around 72.1% of the normalised signal. CHO, CHONS and CHOS groupings comprised around 1% of the total sample signal abundance. These three groups of compounds should not be completely neglected since

the SOA particle mass concentration of this system was appreciable compared to other systems. It might be expected to find a significant contribution of CHO compounds arising from formation of organic acids (e.g. acetyl acrylic acid and glyoxylic acid) under high NOx *o*-cresol photo-oxidation (Schwantes et al., 2017).

Nitro-compounds retaining the carbon number of the parent VOC dominated the CHON grouping. The C$_6$-C$_8$ components were identified as methyl-nitrocatechol, C$_7$H$_7$NO$_4$, isomers (See Table(S1)). OH reaction with o-cresol forms various di-

hydroxytoluene isomers via addition of OH group to different positions on the ring (Olariu et al. (2002).  Subsequent hydrogen abstraction followed by NO$_2$ addition on the ring at the moderate NOx concentrations of our experiments was a likely dominant fate of di-hydroxytoluene in the current study to form the observed dihydroxy nitrotoluene. Further discussion of these isomers is presented in section 3.2.3.1(e).  Schwantes et al. (2017) reported that H abstraction was not the dominant pathway in dihydroxy toluene oxidation, with dihydroxy nitrotoluene only detected at low concentrations by CIMS, with a significant

number of highly oxygenated multi-generational products (mainly CHO compounds) detected by offline direct analysis in real-time mass spectrometry (DART-MS).  It should be noted that the high signal contribution of CHON compounds, dominated by nitro-aromatics in *o*-cresol photo-oxidation (Kitanovski et al., 2012), in Fig. 4(c) may be influenced by their high negative mode sensitivity using electrospray ionisation (Kiontke et al., 2016).


### d) binary α-pinene / isoprene mixture

The binary α-pinene/ isoprene mixture generated considerable particle wall-loss corrected SOA particle mass in all experiments (~101 µg/m$^3$ in the representative one shown here); lower than in the single precursor α-pinene system, but much higher than





in the isoprene system. The distribution of elemental categories of the particle phase products in this system was very similar

to that in the single precursor α-pinene experiments, with CHO compounds dominating the total signal, mainly with between

6 and 10 carbon atoms or between 16 and 20 (Fig.4(d)).   The normalised signal contribution of compounds confidently found

in each repeat in the CHON group was slightly increased in the binary α-pinene/isoprene system (14.4%) compared to single

α-pinene system (10.8%) with a modest enhancement of compounds with greater than 15 carbon atoms (from 3.3% to 6.0%).

In addition, the contribution of large compounds (nC >15) was enhanced in the CHON and CHONS categories in the binary

system compared to single VOC α-pinene system.

This profile is consistent with the domination of the chemical composition in the mixture by α-pinene products, which is

unsurprising since α-pinene is established as a much high yield SOA yield compound than isoprene, especially under neutral

seed conditions (Ahlberg et al., 2017; Eddingsaas et al., 2012; Henry et al., 2012).


**e) binary systems containing o-cresol**

As shown in fig.1(a), the isoprene/*o*-cresol system produces a low particle mass concentration (~22μg/m$^3$) whilst the α-

pinene/*o*-cresol mixture generated the second highest particle wall-loss corrected SOA mass concentration (~150μg/m$^3$).

Compounds found across repeat experiments in these mixtures containing *o*-cresol show the same dominance of CHON signal

as the single precursor *o*-cresol experiment (α-pinene/*o*-Cresol, Isoprene/*o*-Cresol) (Fig 4(e) and 4(f)). The contribution of

CHON compounds to the total SOA increased to approximately 87.8% and 96.0% when α-pinene and isoprene were introduced

into the mixed precursor systems, respectively.  Moreover, the contribution of CHO signal intensity increased in both binary

*o*-cresol mixed systems compared to the single precursor *o*-cresol system. Also, the *o*-cresol / isoprene binary mixture (Fig.4(f))

showed an slightly increased proportion of signal in CHONS compounds at 1.0%. (compared with 0.6% in the single precursor

*o*-cresol system (Fig.4(c)), though noting that the total mass concentration in the mixed system at the end of the experiment

was a factor of 5 lower than in single VOC *o*-cresol system.

The presence of biogenic precursors leads to additional formation of CHO compounds, while the relative signal contribution

of CHON compounds is reduced in each binary system compared to single VOC *o*-cresol system.  A plausible explanation for

this observation could be the increase in O$_3$ generated in the binary mixture, increasing the ozonolysis of 1$^{st}$ generation *o*-cresol

products with double bonds and hence a CHO contribution than in the sole *o*-cresol system.Overall, the negative ionisation

mode signal from the SOA components in a binary mixture containing both biogenic and anthropogenic precursors in our

systems were dominated by categories of components found in the single anthropogenic precursor system, specifically the

CHON group dominated by nitro-aromatics. This may be considered somewhat surprising in the case of the mixture with α-

pinene, since α-pinene (as widely reported and shown in Fig.1) produces higher SOA mass concentration than *o*-cresol under

the same initial conditions as the mixture experiment.





**f) ternary α-pinene / isoprene / o-cresol mixture**

Figure 4 (g) shows the group contribution of the signals in the ternary mixed VOC system corresponding to its moderately
high SOA particles mass concentration (~85μg/m³) shown in fig.1(a). Across the compounds found in all repeat experiments,
whilst not as completely dominant as in the *o*-cresol containing binary systems, the substantial (65.7%) C$_6$-C$_8$ CHON
contribution again shows that the *o*-cresol-derived nitrocatechols play a significant role. CHO compounds make a significant
contribution with normalized abundance ~14%. Whilst the CHON compounds mainly consist of C$_6$-C$_8$ compounds, the CHO
compounds comprise both C$_6$-C$_8$ and C$_9$-C$_{10}$ compounds. SOA production in the ternary system appears not to be entirely
driven by any single precursor and additionally, the overwhelming negative mode CHON dominance which may be controlled
by sensitivity of the electrospray method, does not appear to the same degree in the ternary system as it does in the *o*-cresol-
containing binaries.

There was a small contribution to the CHO group from compounds with more than 15 C atoms. Whilst relatively low in
normalised signal contribution, they were found in all ternary repeat experiments and can be presumed to be accretion products.
As an indication of the relative contribution of accretion products to the SOA particle mass in each system, table S2 shows the
signal-attributed mass concentration of molecules with nC>21 that observed confidently in all repeat experiments, by scaling
the fractional signal contribution to the measured PM mass at the end of the experiment. The signal-attributed mass
concentration of these large molecules is around 6, 575 and 80 times lower in the single VOC isoprene system (0.002μg/m³)
than in the isoprene / *o*-cresol (0.013 μg/m³), α-pinene / isoprene (1.15 μg/m³) and ternary (0.16 μg/m³) mixtures respectively.

**3.2.2.2. Negative Ionisation Aggregate Particle Component Properties**

This section describes average properties of the SOA PM mass using a variety of chemical metrics including molar carbon
number (nC), molar hydrogen to carbon ratio (H/C), oxygen to carbon ratio (O/C), average oxidation state ($\overline{OSc}$), double bond
equivalent (DBE) and double bond equivalent to carbon ratio (DBE/C). The molar carbon number reflects to the average size
of SOA particle components and often the major condensed-phase products retain the same carbon number as the precursor
(Romonosky et al., 2015). The H/C and O/C provides summary information about chemical composition of bulk organics, and
OSc corresponds to the average degree of oxidation of carbon in the organic species (value of $\overline{OSc}$ increasing upon
oxidation)(Daumit et al., 2013; Safieddine and Heald, 2017). The OSc values were calculated by using 2*O/C-H/C for CHO,
CHONS, and CHOS compounds due to the low measured abundances fractions of two species in the oxidation products we
observed in section 3.2.2.1 and 3.2.2.3. For CHON compound, equation $\overline{OSc}$ =2*O/C-H/C-(OS$_N$*N/C) was used to determined
the $\overline{OSc}$. The OS$_N$=+5 if nO>=3 and OS$_N$=+3 if nO <3 for CHON compounds (Kroll et al., 2011). It is common to used DBE
and DBE/C to quantify the unsaturated bonds (and aromaticity) in a molecule. The DBE corresponds to the sum of unsaturated
bonds (including aromatic and cycloalkene ring) and increasing DBE/C ratios indicates increasing contribution of the signal
from molecules containing aromatic rings (Koch and Dittmar, 2006).





Table 3 shows the signal-weighted chemical metrics from compounds detected in all repeat experiments in each system. All properties were normalised to the total detected compound abundance. All parameters in the single VOC α-pinene and binary α-pinene/isoprene systems are similar, consistent with the dominance of α-pinene-derived particle mass in the binary system. In contrast, the H/C value decrease from 1.46 to 1.03 and the O/C value remain constant (~0.5) in binary α-pinene /o-cresol

compared to the single VOC α-pinene system. Indeed, the signal-intensity weighted average values of all chemical parameters shows that the o-cresol single VOC system aggregate properties are very similar to those in both o-cresol containing binary system, with an understandably high level of aromaticity (DBE/C >=0.67) (Koch and Dittmar, 2006), indicating that oxidation and partitioning to the particles in the unary and binary o-cresol systems is largely ring preserving. The OSc value decreased from -0.55 to -0.63 in α-pinene / o-cresol compared to the single VOC o-cresol system suggests less oxidised products were

formed when introducing α-pinene precursors into single o-cresol system. The abundance-weighted average values of all chemical parameters in the particles in the ternary mixture do not show common features with any single precursors system, with the coincidental exception of the nC and O/C value that are similar to that in the o-cresol system.

**Table 3: Intensity Weighted Average Values from negative ionization mode LC-Orbitrap MS for O/C, H/C, Osc, DBE/C, DBE and**
**the number of carbons present (nC) for SOA filters extracts from single and mixed precursor experiments**

| Chemical parameters | α-pinene | Isoprene | o-cresol | α-pinene/ Isoprene | Isoprene/ o-cresol | α-pinene/ o-cresol | α-pinene/ Isoprene/ o-cresol |
|---|---|---|---|---|---|---|---|
| nC | 11.57 | 7.08 | 7.01 | 10.63 | 7.03 | 7.56 | 7.75 |
| H/C | 1.46 | 1.27 | 0.99 | 1.46 | 1.00 | 1.03 | 1.08 |
| O/C | 0.51 | 0.81 | 0.57 | 0.52 | 0.48 | 0.52 | 0.55 |
| $\overline{OSc}$ | -0.58 | 0.26 | -0.55 | -0.57 | -0.7 | -0.63 | -0.55 |
| DBE/C | 0.39 | 0.57 | 0.71 | 0.40 | 0.70 | 0.68 | 0.65 |
| DBE | 4.28 | 3.90 | 5.01 | 4.37 | 4.98 | 5.02 | 4.89 |

The weighted average number of carbons in α-pinene experiment (~11) indicated that a modest accretion reaction (including oligomerization and functionalization) occurred in oxidation, and that the α-pinene particle phase oxidation products had significant impact on α-pinene / Isoprene binary system. The average carbon number of isoprene SOA particles was larger

than the isoprene precursor ($C_5$), implying particle-phase accretion reactions such as organosulfate formation though forming very little particle mass in the current study. The similarity of properties between the single VOC o-cresol system and its binary mixtures suggest that common compounds dominate the signals, and from Fig.4(c) , (e) and (f) it can be seen that these are compounds in the CHON elemental category. In addition, the DBE/C values indicate dominance of the major oxidation products in these o-cresol containing systems by condensed aromatic structure, consistent with the finding in Ahlberg et al.

625 (2017).





### 3.2.2.3. Positive Ionization mode

Figure 5 presents the positive ionisation mode signal fraction of compounds in representative experiments that can be confidently stated as found in each of the systems (i.e. found in every repeat experiment in this system) in the coloured stacked

bars according to their carbon number and classified according to their elemental CHO, CHON, CHOS, and CHONS categories. Also shown in the grey bar is the signal fraction of compounds that are inconclusively found in the experiment in each system classified by elemental grouping, but not found in all repeat experiments and chamber background. The fractional contributions of confidently stated as products are similar for each experiment in a particular system.

It is evident that there is a generally a greater fraction of the positive ionisation mode signal that is inconclusive than in negative

ionisation mode as shown in Figure 4. This indicates a larger variability in composition between repeat experiments with some compounds not found in some repeats experiments, or a larger fraction of the signal from compounds also found on chamber background filters.


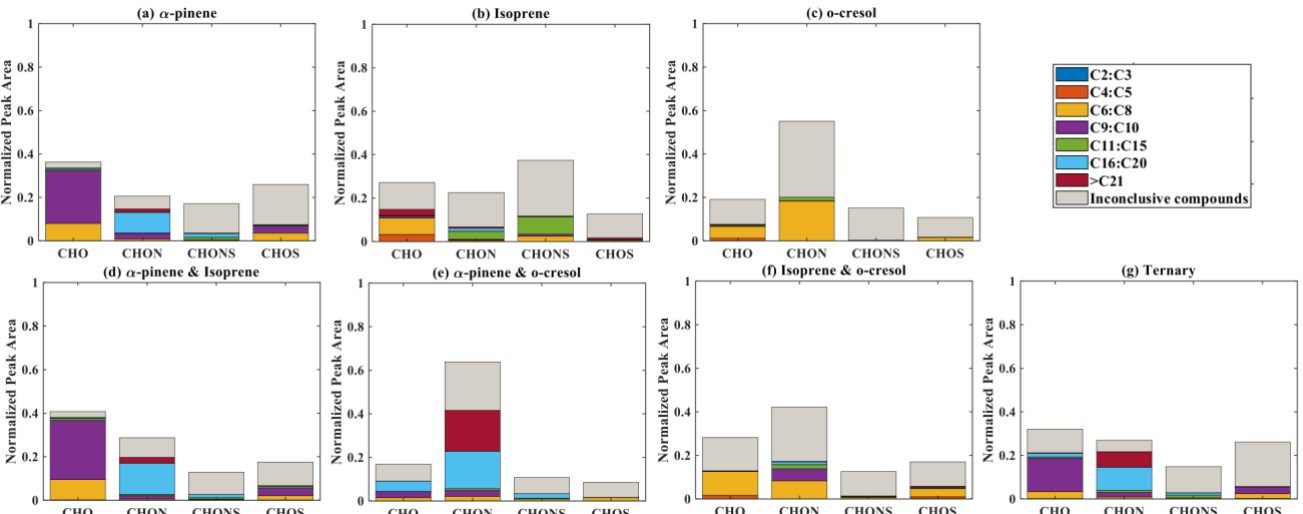

**Figure 5: The normalized signal intensity distribution of different compound categories (CHO, CHON, CHOS, and CHONS) for various single and mixed precursor systems in Positive ionization mode by LC-Orbitrap MS. The grey bar (inconclusive compounds) signal attributed to compounds that were not universally found in all repear experiments.**


### a) α-pinene

In the single precursor α-pinene system (Fig 5. (a)), 33.5% of the total signal abundance was from CHO compounds found in each repeat experiment with the majority of molecules containing between 6 and 10 carbon atoms. The compounds confidently





found in the CHOS category provided 7.4% of the signal fraction, also mainly comprising compounds with 6 to 10 carbon
atoms.  The remainder of signal was observed in CHON (14.6%) and CHONS (3.5%) categories, which were found in all
repeat experiments in this system mainly comprised large compounds, with some nC<11 molecules in these CHON category.
The contribution of $C_9$ to $C_{10}$ molecules in the CHO and CHONS categories are consistent with previous studies of α-pinene
ozonolysis and OH oxidation in the presence of NOx and seed particles (Winterhalter et al., 2003, Yasmeen et al., 2012).  The
signal contribution of CHO compounds and CHOS with carbon number 6 to 8 suggested that fragmentation plays an important
role. It is likely that these compounds formed from fragmentation of alkoxy radicals ($RO_2$+ NO→RO+$NO_2$)(Pullinen et al.,
2020).

The CHOS and CHONS compounds may be attributed to esterification of α-pinene SOA.  Experimental results from Surratt
et al. (2007b) reported that  the sulphate ester and/or its derivatives has significant contribution  in SOA formation of α-pinene
photo-oxidation in the presence of ammonium sulphate seed. The large molecules in CHON and CHONS groups suggest the
occurrence of accretion reactions between peroxy-peroxy radicals containing nitrogen and sulfur ($RO_2$+ R'$O_2$→ROOR'+$O_2$)
(Pullinen et al., 2020).

### b) Isoprene

Considering only the compounds found in all repeat experiments and not on the background filter, the dominant contribution
in the isoprene signal in its single VOC photo-oxidation system were observed in the CHO category with a normalized signal
of 14.8%, with molecules mostly comprising 5-8 carbon atoms or larger molecules with carbon number greater than 9 (Fig
5(b)). CHONS compounds are the next largest constituent, with 11.7% of the total signal.  More than half of the CHONS signal
is from large molecules (nC>11), the rest of the CHONS compounds mainly comprising molecules with carbon number 6 to
10.  The remainder of the signal was found in the CHON and CHOS categories with carbon number greater than 11.
Compounds which could not be confidently attributed to the isoprene system owing to their sole presence in every repeat
experiment made a significant contribution (~65%)  to the total signal, though an even greater fraction of inconclusive signal
(78.9%) was observed in negative ionization mode (likely resulting from the extremely low total mass yield).

Clearly accretion reactions dominated the isoprene system in the positive (as well as negative) ionisation modes.  Contribution
of CHONS compounds in total SOA are consistent with the formation of organosulphate and nitroxy organosulphate by
uptake of isoprene oxides on ammonium sulphate particles (Surratt et al., 2007b; Surratt et al., 2007a). Moreover, the presence
of $C_4$-$C_5$ molecules in CHO categories could be simply explained by the gas phase oxidation pathway of isoprene, though as
with the negative mode samples, it is unclear why such small molecules partition to the particle phase.  The possible
interpretations is that weakly bound large molecules fragment during LC-Orbitrap MS analysis, and /or due to the possibly gas
phase filter absorption.





### c) *o*-cresol

Fig.5c shows that approximately 20% of the signal is in the CHON category in the single VOC *o*-cresol system. The majority (18.3%) of signal from confidently attributable molecules found in all repeat experiments in this CHON category contain 6 to 8 carbon atoms. Signal in the CHO, and CHOS categories is similarly dominated by compounds containing 6 to 8 carbon atoms with fractional contributions of 5.4%, and 1.7% respectively. Specifically, the $C_7$ compounds has fractional signal contributions 3.8%, which is approximate 3 times higher than $C_6$ (1.1%) and about 12 times higher than $C_8$ molecules (0.3%)

in CHO categories. In the CHOS categories, $C_6$ organic species (1.3%) made the dominant contribution compared to $C_7$ (0.3%) and $C_8$ (0.05%) species. The CHONS category in this system almost entirely comprised molecules that were not found in all repeat experiments so are considered inconclusive in this analysis.

Compounds found in all repeat experiments with between 11 and 15 carbon atoms in the CHON category account for 1.6 % of the signal (with $C_{14}$ 1.5% and $C_{11-12}$ 0.1%). It is likely that the majority of of $C_{11}$ to $C_{15}$ signal is attributed to $C_7$ dimers.


$C_{6-8}$ CHON compounds are likely to be second-generation *o*-cresol oxidation products such as dihydroxy nitrotoluene which are also detected in negative ionization mode as a result of being both protonated and deprotonated. The CHO compound present in this study might have some contribution from multi-generation products generated from decomposition of bicyclic intermediate compounds formed from OH oxidation of o-cresol, as reported by (Schwantes et al., 2017), but are probably

mainly dihydroxy toluene compounds, which have been reported with a 70% yield from *o*-cresol oxidation (Olariu et al., 2002). Decomposition of bicyclic intermediate compounds leading to formation of unsaturated carbonyl molecules could form oligomeric species resulting in formation of the $C_{11-15}$ molecules in the CHO, and CHON groups.

### d) binary α-pinene / isoprene mixture

The elemental categories in the binary α-pinene/isoprene samples shown in Fig.5(d) indicate a high similarity to the single VOC α-pinene system (Fig.5 (a)), with CHO compounds dominating the total signal and predominantly containing 9 to 10 carbon atoms, but with some fragmentation to $C_6$-$C_8$. The signal intensity of CHOS compounds reduced by 0.7% of the total signal in the binary system (Fig 5. (d)) compared to the single VOC α-pinene system (Fig 5.(a)), mostly in the $C_6$-$C_8$ signal. In contrast, the signal intensity of CHON components is 19.7% of the total in the binary system, 5.1% higher than in the single

VOC α-pinene system, with the enhancement in molecules with carbon number >16.

The similarity in the elemental categorisation between single VOC α-pinene and binary α-pinene / isoprene system again supports the contention that α-pinene components dominate the total signal in the binary system. However, the enhancement of CHON compounds intensity in binary system possible implies an increase in the $RO_2/NO_2$ or $RO_2/NO$ termination pathways

leading to stronger organic nitrate formation. A large fraction of the signal from molecules with carbon number greater than 16 in this binary system might be attributed to dimerization of gas-phase nitrated highly oxidized molecules.



**e) binary systems containing o-cresol**

The distribution of SOA products from α-pinene / *o*-cresol (Fig.5(e)) and isoprene / *o*-cresol binary systems (Fig 5(f)) show

obvious differences compared to the corresponding single precursor systems. In the α-pinene / cresol binary system, the dominant signal intensity was contributed by CHON compounds, and they mainly comprise molecules with more than 16 carbon atoms. The rest of the signal was found in the CHO (9.0%), CHOS (1.6%), and CHONS (3.3%), categories, while compound with nC>=9 made up a significant proportion.  In the isoprene/*o*-cresol system, most of the compounds were in the CHON category (17.2%), and the majority of them were composed of 6 to 10 carbon atoms.  $C_9$-$C_{15}$ molecules also made a

non-negligible contribution in CHON compounds (7.4%). The remainder of the signal was found in the CHO (12.9%), CHONS (1.3%) and CHOS (5.8%) categories, again concentrated at $C_{6-8}$.

Lack of similarities between *o*-cresol containing binary systems and the corresponding sole precursor systems in the positive ionisation mode, suggests a significant contribution to the signal from the unique compounds shown in Figure 3 exerting some

control over the elemental composition of SOA in binary systems. For instance, cross-products from α-pinene and *o*-cresol gas or particle-phase oxidation probably contribute to the high carbon number compounds in binary system.  In the isoprene/*o*-cresol system, high $C_6$-$C_8$ contributions in all categories were likely from *o*-cresol, though the other contributions were dissimilar to the individual precursor systems.

**f) ternary α-pinene / isoprene / o-cresol mixture**

The distribution of SOA products in ternary system (Fig.5(g)) was very similar to the single precursor α-pinene experiments (Fig.5(a)). The dominant compounds were found in the CHO categories with signal intensity of 21.1%, most of them with 6 to 10 carbon atoms.  The 17.5% signal contribution of molecules with carbon number greater than $C_{16}$ in CHON is 7.2% higher than signal intensity of CHON molecules with carbon number >16 in the single precursor α-pinene system (10.3%).

The most notable difference between the positive mode signal in the ternary system and the single precursor systems was the high contribution of molecules with nC > 21 in the CHON category. As an indication of the relative contribution of accretion products to the SOA particle mass in each system, table.S2 shows that the signal-attributed mass concentration of molecules (nC>21) in the single VOC isoprene system, at 0.016 $\mu$gm$^{-3}$, is significant lower than in the α-pinene/o-Cresol binary (2.85 $\mu$gm$^{-3}$) , and is about 8 times less than in the isoprene / *o*-cresol binary (0.14 $\mu$gm$^{-3}$) and 70 times less than the ternary (1.10

$\mu$gm$^{-3}$) systems, which is comparable to the single precursor α-pinene system (1.34 $\mu$gm$^{-3}$).  The SOA particle products of the ternary system are mainly attributable to α-pinene oxidation, and accretion reactions, possibly across different precursor products, leading to high carbon number nitrogen-containing compounds.





### 3.2.2.4. Positive Ionisation Aggregate Particle Component Properties

Table 4 shows the intensity weighted average values for compounds detected in positive ionisation mode in all repeat experiments of individual SOA systems. All properties were normalised to the total detected compounds abundance. Clearly, the nC values in all three single VOCs systems were higher than their precursor's carbon number. For example, the nC values in isoprene SOA is 11.73, which is 2 time higher than carbon number of isoprene ($C_5$). In the binary α-pinene/Isoprene system, the nC (11.90) slightly higher than in single α-pinene system (11.55), and in single isoprene system, suggesting a contribution from each. The OSc values seem comparable in both single systems and binary α-pinene/isoprene system. The average value of nC in binary α-pinene/o-cresol system (17.88) was significantly higher than the single VOC α-pinene (11.55) and o-Cresol systems (7.61). The O/C values in binary α-pinene/o-cresol system was approximately 0.15 lower than sole α-pinene and o-cresol system, while the H/C values in binary α-pinene/o-cresol system is comparable to single α-pinene, and about ~0.4 higher than sole o-cresol system. The average value of nC in the binary isoprene/o-cresol system (8.43) was lower than sole isoprene systems (nC=11.73), but higher than single o-cresol system (nC=7.61). The signal intensity weighted values for all chemical parameters in the ternary mixture shown no obvious similarity to those in any sole precursor system with the exception of DBE/C parameter.

It is apparent in the positive mode that accretion reactions occurred, and its products play an essential role in single isoprene system, the binary α-pinene-containing systems and the ternary system. Moreover, although some of the chemical parameters in the binary system show similar values compared to single precursor systems, the significant differences between mixed systems and those of the individual precursors imply that categories of components in the mixed systems were controlled by the compounds that were unique to the mixture and not found in the single precursor systems.

**Table 4: Intensity Weighted Average Values obtained from positive ionization mode LC-Orbitrap MS for O/C, H/C, OSc, DBE, and the number of carbons (nC) present for SOA filters extracts from single and mixed precursor's experiment.**

| Chemical parameters | α-pinene | Isoprene | o-Cresol | α-pinene/ Isoprene | Isoprene/ o-Cresol | α-pinene/ o-Cresol | α-pinene/ Isoprene/ o-Cresol |
|---|---|---|---|---|---|---|---|
| nC | 11.55 | 11.73 | 7.61 | 11.90 | 8.43 | 17.88 | 13.69 |
| H/C | 1.56 | 1.65 | 1.09 | 1.54 | 1.25 | 1.55 | 1.52 |
| O/C | 0.32 | 0.36 | 0.36 | 0.29 | 0.46 | 0.17 | 0.23 |
| $\overline{OSc}$ | -0.95 | -1.00 | -0.66 | -1.03 | -0.50 | -1.38 | -1.17 |
| DBE/C | 0.32 | 0.32 | 0.64 | 0.33 | 0.54 | 0.31 | 0.33 |
| DBE | 3.72 | 3.15 | 4.82 | 3.86 | 4.28 | 5.49 | 4.55 |



### 3.2.2.5. Insights from the combination of positive and negative mode elemental categorisation of signal contribution

Considering the results of both negative and positive ionization modes, the α-pinene derived compounds unsurprisingly dominate the elemental categorisation of binary α-pinene /isoprene binary system, since the α-pinene produced a much greater mass concentration than isoprene. The average carbon number in positive ionization mode (Table 3 and 4) shows that SOA formation in the binary α-pinene / isoprene binary system involves similar accretion products as found in the single VOC α-pinene system. Whilst *o*-cresol generated appreciable SOA particle mass concentration, this was still significantly lower than in the single α-pinene system. However, the negative mode analysis suggests that *o*-cresol oxidation products can make a more significant contribution than α-pinene products, notwithstanding the particularly high sensitivity to aromatic nitro-compounds, which make a high contribution to the *o*-cresol CHON category. The positive mode, being sensitive to a different subset of the compounds, differs to the observations in negative ionisation mode. The observation in positive mode reveals that SOA elemental composition in binary *o*-cresol/ α-pinene system is not driven by any single precursors' oxidation products but by the new compounds that appear to be *o*-cresol / α-pinene large molecular cross-products. Approximately half of number of compounds were unique in the binary *o*-cresol/ α-pinene system in both positive and negative modes (Fig. 2(b) and Fig 3(b)). In the *o*-cresol/isoprene system, it may be expected that the elemental composition was driven by the *o*-cresol since the isoprene oxidation produced very little particulate mass compared to that of *o*-cresol. Negative ionization results were consistent with this, though positive mode indicated an additional significant contribution from *o*-cresol / isoprene large molecular cross-products. Overall SOA particle formation in binary systems can be seen to be mainly dependent on the high yield precursors, but also influenced by the interaction between products of the individual precursors, with the unique compounds making a greater contribution than any sole precursor's products in positive ionization mode of the *o*-cresol/ α-pinene system. In the ternary system, the elemental composition shares striking resemblance of single α-pinene system in positive ionization mode, but in negative ionization mode, there was little similarity with any single precursor system, with all three precursors contributing. On the other hand, the elemental grouping results clearly shown that the compounds that were not present in all repeat experiments and hence inconclusively attributable in all precursor system made non-negligible contributions in both modes (especially in positive ionization), suggesting the repeatability of SOA chemical composition in each system is not ideal. This may be an artefact of the inhernt difficulty of precisly replicating operating process during chamber experiments. It should not affect the analysis of SOA chemical characterization between single and mixture precursor systems, since only the confidently attributable compounds between repeat experiments were employed for comparison.





### 3.2.3.    Molecular Characterization of Particulates Organics

### 3.2.3.1.   Negative Ionization Mode

810   This section aims to investigate whether the components in mixtures were also present at significant fractional abundance in particles or absent from any of the single VOC photo-oxidation system. The absence in the single VOC systems of those components making a substantial contribution to the mixtures may be indicative of interactions during the photochemistry and multiphase processing giving rise to tracers of the combinations of VOC precursors in multicomponent particles that may be of use in SOA source attribution in future ambient studies. The normalized peak area of 15 selected compounds in binary

815   mixed system and 20 selected compounds ternary mixed system are shown in Figure 6. In all mixed systems, the five compounds with the highest signal fraction that were also present in each corresponding single precursor systems are shown alongside the top five compounds uniquely found in the mixture but absent from any single precursor system.

Only compounds found in all repeat experiments in each system were chosen for this analysis, so there is confidence in the

820   component identification.

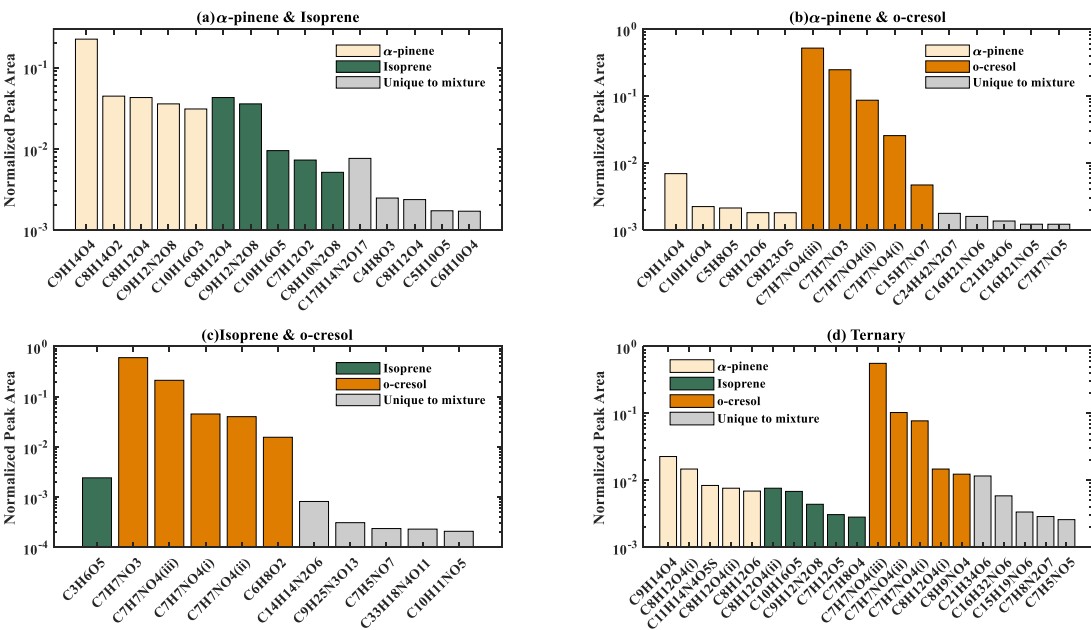

**Figure 6:The dominant 15 compounds in terms of their normalized peak area in the mixed VOC systems shown in the bars: (a) binary α-pinene / Isoprene system; (b) binary α-pinene / _o_-cresol; (c) binary Isoprene / _o_-cresol; (d) ternary system respectively.**

825   **The normalized peak area of these selected 10 compounds in a mixed precursor's system are also presented if they existed in corresponding single precursor's system (yellow: single α-pinene, green: isoprene, orange:o-cresol) . The compounds are considered**





**identical in the mixed system and single VOC systems if they have the same empirical formula and a retention time difference of <0.1min in negative ionization mode.**

#### a) the binary α-pinene / isoprene system

The components in binary mixture system that were also found in the single precursor α-pinene system were found to have a larger signal fraction than those found in the single isoprene system and the unique compounds. In particular, it was found that $C_9H_{14}O_4$ made the greatest signal contribution (Fig6(a)). This is also the case in the single precursor α-pinene system, and is likely to be pinic acid due to this peak has similar fragmentation pattern (Figure S3) compared to results reported in Yasmeen et al. (2010). $C_8H_{14}O_2$ and $C_9H_{12}N_2O_8$ made a non-negligible contribution in the binary mixture system with normalized molecular abundance 4.4% and 3.4% and were also found conserved in both single precursors systems. Compounds that were only present in the binary mixture had relatively low abundance, with the highest contribution from $C_{17}H_{14}N_2O_{17}$ with only 0.7% of total signal fraction. Clearly the SOA particle composition in the binary α-pinene / isoprene system was dominated by α-pinene components and partially contributed by isoprene, but not those from cross-products from their interaction.

#### b) binary α-pinene / o-cresol system

As shown in (Fig.6 (b)), the most four abundant peaks (three $C_7H_7NO_4$ isomers (i to iii) and $C_7H_7NO_3$) in the mixture were found to be present in the single precursor o-cresol system. $C_7H_7NO_4$ isomers in the binary mixture had signal contributions of 51.7% (iii), 8.6%(ii) and 2.5% (i) respectively, with the $C_7H_7NO_3$ contribution 24.3%. $C_9H_{14}O_4$ is present in both the single VOC α-pinene system and mixed system, with relatively high (0.68%) signal contribution in the mixed system compared to the other four compounds common to the mixture and α-pinene alone. The top five unique compounds in binary mixture system had small normalized signal fraction to total sample abundance in a range of 0.12% to 0.17%.

The four dominant compounds in the binary mixture are all nitro-aromatic compounds formed in the oxidation of o-cresol (Schwantes et al., 2017; Kitanovski et al., 2012). $C_7H_7NO_4$ are multiple isomers of methyl-nitrocatechol with the methyl, hydroxyl and nitro groups at various positions on the aromatic rings. $C_7H_7NO_3$ was identified as methyl-nitrophenol. (Details of deprotonated species of $C_7H_7NO_4$ and $C_7H_7NO_3$ in table S1). As with the group categorisation, care must be taken with the interpretation of the molecular contributions to the signal owing to the enhanced sensitivity of electrospray ionisation.

#### c) binary isoprene / o-cresol system

Fig. 6(c) shows that only one compounds in the binary isoprene / o-cresol system was unequivocally observed in all repeat experiments the single isoprene precursor system. Components present in single o-cresol system make higher contribution in binary mixture system than isoprene derived compounds and those unique to the mixture, with one $C_7H_7NO_3$ and three $C_7H_7NO_4$ isomers making the most significant contribution. According to the deprotonated molecular species fragmentation (table S1), three $C_7H_7NO_4$ isomers were found at retention time 9.14, 4.52 and 7.53. These three $C_7H_7NO_4$ have similar





fragmentation ions that relate to loss of NO ion (m/z=138) and NOH ion (m/z=137). The five compounds that were unique to the mixture were found to make negligible contribution to total sample abundance (between 0.05% to 0.2%).

As in the α-pinene / o-cresol binary mixture, the compounds found in the *o*-cresol system dominate the SOA particles in the binary isoprene / o-cresol system. Isoprene-derived compounds were found to make a negligible contribution; all of dominant compounds in the binary system were found in the single VOC *o*-cresol system, and only one compounds in the single VOC

isoprene system. There is no evidence to suggest that a compound has high enough contribution to act as tracer for the binary mixture. The three dominant compounds ($C_7H_7NO_4$ isomers) were uniquely identified as *o*-cresol oxidation products (methyl-nitrocatechol isomers) with similar retention time and fragmentation ions as the $C_7H_7NO_4$ compounds that were found in the binary α-pinene / o-cresol system. As with the group categorisation the consideration of enhanced sensitivity of electrospray ionization must be borne in mind in the isoprene/o-cresol and α-pinene / *o*-cresol mixtures.


### d) the ternary mixture

In the ternary system (Fig.6 (d)), the top 3 largest contributing signal ($C_7H_7NO_4$ isomers) are an *o*-cresol oxidation product, the other two o-cresol compounds has comparable normalized peak area (~1.2%). Also, the α-pinene SOA also make non-negligible contribution in a range of 0.7% to 2.2%  in the ternary mixture system though significantly lower than o-cresol

derived compounds. Five isoprene derived compounds (0.28% to 0.76%) make comparable signal contribution to the five unique compounds (0.25% to 1.1%).

*o*-cresol SOA and α-pinene SOA clearly significantly influenced the chemical composition in the ternary system, while the isoprene SOA and unique compound contributions are modest. A unique potential tracer compounds ($C_{21}H_{34}O_6$) was only

observed in this ternary mixture of α-pinene, *o*-cresol and isoprene with a 1.1% contribution and was found in all repeat experiments.

### 3.2.3.2.  Positive Ionization Mode

Figure 7 compares the normalized peak area of selected compounds in mixed and single precursor systems in Positive

Ionization Mode.  As the negative ionization mode, figure 7 shows 15 selected compounds in each binary mixed system and 20 compounds in ternary mixed system, following the same selection criteria.





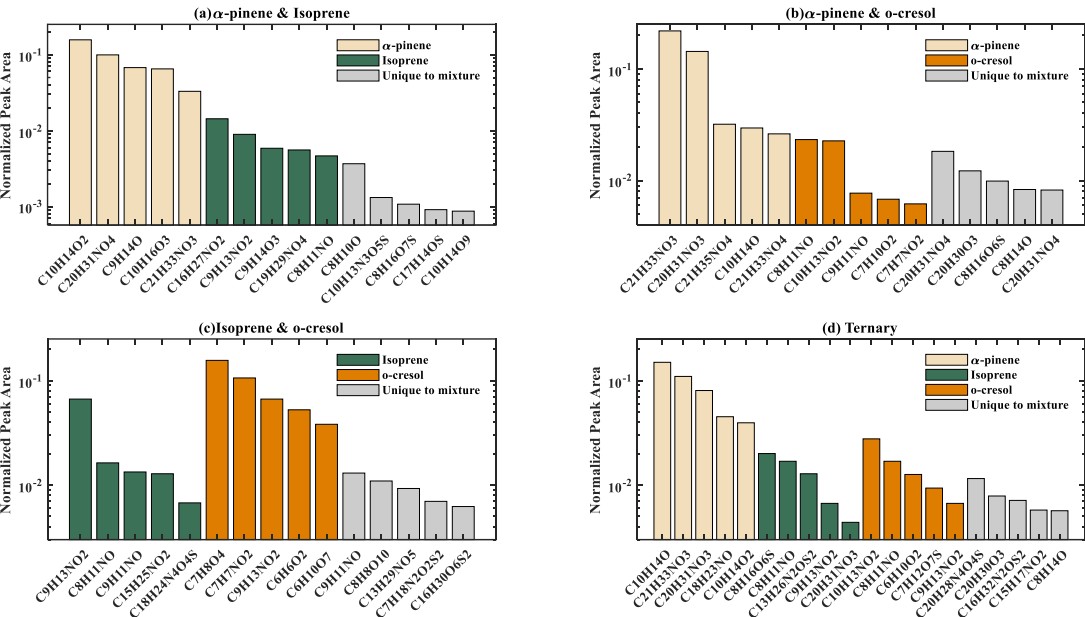

**Figure 7:The dominant 15 compounds in terms of their normalized peak area in the mixed VOC systems shown in the bars: (a)**
**binary α-pinene / Isoprene system; (b) binary α-pinene / *o*-cresol; (c) binary Isoprene / *o*-cresol; (d) ternary system respectively.**
**The normalized peak area of these selected 10 compounds in a mixed precursor's system are also presented if they existed in**
**corresponding single precursor's system (yellow: single α-pinene, green: isoprene, orange:o-cresol) . The compounds are considered**
**identical in the mixed system and single VOC systems if they have the same empirical formula and a retention time difference of**
**<0.1min in positive ionization mode.**


### a) the binary α-pinene / Isoprene system

Fig.7(a) indicates that α-pinene derived compounds dominated the binary α-pinene / isoprene system, $C_{10}H_{14}O_2$ with the

highest normalized peak area of 15.5% followed by $C_{20}H_{31}NO_4$ at 8.6%. The contribution of isoprene derived compounds

(0.5%-1.4%) is lower than that of those derived from α-pinene, but higher than compounds unique to the mixture, the highest

fractional abundance of which was 0.32% ($C_8H_{10}O$).

The particle components in the binary α-pinene/isoprene system was substantially driven by the α-pinene components as found

in negative ionization mode, likely resulting from the low SOA yield of isoprene oxidation under the conditions of our

experiment. The isoprene components had little influence on the composition in this system. There is insufficient information

to suggest that a compound has high enough contribution to act as tracer for the binary mixture, but compounds unique to this

mixture with seed particles under moderate NOx conditions were found to be sulphur-containing.





### b) binary α-pinene / o-cresol system

Two α-pinene derived compounds dominated this system ($C_{21}N_{33}NO_3$ and $C_{20}H_{31}NO_3$) in positive ionisation mode (Fig.7 (b)). The other three α-pinene derived compounds were found at comparable levels to the top two derived from o-cresol ($C_8H_{11}NO$ and $C_{10}H_{13}NO_2$) at approximately 2.5% of total molecular signal. The contribution of compounds unique to the mixture were lower than all five α-pinene derived compounds, but higher than most of o-cresol SOA. The highest contribution from these unique compounds was $C_{21}H_{33}NO_4$ with 1.8% signal intensity.


Although both α-pinene and o-cresol oxidation products contributed to this system, the most abundant peaks ($C_{21}H_{33}NO_3$ and $C_{20}H_{31}NO_3$) were only found in the single precursor α-pinene system but not in the single precursor o-cresol system. The nitrogen-containg compound ($C_{21}H_{33}NO_4$) might act as tracer compound for binary system, which is possibly driven by further oxidation of $C_{21}H_{33}NO_3$ compound.


### c) binary isoprene / o-cresol system

Fig 7(c) shows o-cresol derived compounds controlled the particulate chemical compositon in this system.The fractional contribution of $C_7H_8O_4$ and $C_7H_7NO_2$ from the o-cresol system were 15.5% and 10.5% respectively in the binary mixture. One isoprene derived compound ($C_9H_{13}NO_2$) made considerable contribution (6.6%) in the binary mixture system. Compounds
unique to the mixture were $C_9H_{11}NO$ (1.3%), $C_8H_8O_{10}$(1.0%), $C_{13}H_{29}NO_5$ (0.9%), $C_7H_{18}N_2O_2S_2$ (0.7%), and $C_{16}H_{30}O_6S_2$ (0.6%).

The higher yield o-cresol made a much more significant contribution to the SOA components than the lower yield isoprene. The significant abundance of two unique compounds ($C_9H_{11}NO$, and $C_8H_8O_{10}$ ) may result from interactions in the mixture
and their exploration for use as tracers of the mixed system might prove useful.

### d) The ternary mixture

From Fig. 7(d), the dominant compounds of the ternary system in positive ionisation mode were derived from α-pinene with fractional contribution range of 0.39% to 14.9%. The highest peak was $C_{10}H_{14}O$ with signal intensity 14.9%. The top o-cresol
derived compounds were $C_{10}H_{13}NO_2$, $C_8H_{11}NO$ and $C_6H_{10}O_2$ with 2.7%, 1.6% and 1.2% signal intensity respectively. The top three isoprene derived compounds were $C_8H_{16}O_6S$ (2.0%), $C_{13}H_{26}N_2OS_2$ (1.6%)and $C_{13}H_{26}N_2OS_2$ (1.2%) respectively. $C_{20}H_{28}N_4O_4S$ was unique to the ternary mixture with a fractional contribution approximately ~1.1%, which could be the products from dimerization of α-pinene products or from interactions in the mixture.

The highest SOA yield α-pinene clearly dominated the product distribution of the ternary mixture in positive ionization mode.
Isoprene and o-cresol derived and unique-to-the-mixture components made little contribution.



## 4. Conclusion

In this study, the SOA chemical composition formed from the photooxidation of α-pinene, isoprene, o-cresol and their binary and ternary mixtures in the presence of NOx and ammonium sulphate seed particles was determined by non-targeted LC-Orbitrap MS. SOA particle mass from isoprene was almost negligible under our experimental conditions, o-cresol generated more and α-pinene the highest, and exhibited the highest yield in our experiments.

The number of detected SOA compounds and their molecular composition indicated that α-pinene oxidation products have a dominant influence on the SOA particle composition in the binary α-pinene/isoprene system, which can involve oligomerization/accretion reactions forming products such as $C_{20}H_{31}NO_4$. The major products in this system shows that SOA composition is clearly driven by the high α-pinene yield with isoprene oxidation products observed to make a minor contribution. The nitrogen containing compound $C_{17}H_{14}N_2O_7$ might be a potential tracer in binary α-pinene/ isoprene system in the presence of ammonium sulphate seed.

The compositional analysis in negative ionization mode reveals that o-cresol products dominate SOA particle composition in the α-pinene/o-cresol system, with major contributions from methyl-nitrocatechol isomers ($C_7H_7NO_4$) and methyl-nitrophenol ($C_7H_7NO_3$), though this will be influenced by the high sensitivity in the employed electrospray ionization method. There is a relatively high contribution to the elemental composition from unique-to-mixture products in positive ionization mode, indicating the significant prevalence of interactions between the oxidation products in this system. The molecular analysis in both ionization modes also indicated that both α-pinene and o-cresol influenced the product distribution in their binary mixture. Similarly, o-cresol oxidation heavily influenced SOA particle composition in the binary isoprene/o-cresol system in negative ionization mode, but unique-to-mixture products had made considerable contribution in the positive ionization mode. The molecular analysis in both modes suggested that higher yield o-cresol products were present in greater abundance than those from isoprene. Two unique compounds ($C_9H_{11}NO$, and $C_8H_8O_{10}$) in positive mode were identified that could behave as tracers in this system.

SOA composition in binary mixtures was therefore generally strongly determined by the oxidation products of the higher yield precursors, but interactions leading to cross-product formation also play an important role, especially in o-cresol containing systems.

In the ternary system, the elemental category composition analysis presented in positive ionization mode suggested that the chemical composition of SOA strongly depends on the sole α-pinene oxidation, with products from the oxidation of α-pinene and o-cresol identified as important in negative ionization mode. The molecular analysis shows that products from both α-pinene and o-cresol strongly influence the composition of SOA particles with very few isoprene oxidation products making a major contribution, indicating a limited role for isoprene oxidation. Moreover, cross-products $C_{21}H_{34}O_6$ and $C_{20}H_{28}N_4O_4S$ were identified as potential tracers in the ternary system.




## Data availability

All the data used in this work can be accessed on the open database of the EUROCHAMP programme (https://data.eurochamp.org/data-access/chamber-experiments/).

## Competing interests

The authors declare that they have no conflict of interest.

## Author contributions

GM, MRA, AV, YW and YS conceived the study. AV, YW, YS and MD conducted the experiments. KP provided on-site LC-
Orbitrap MS training for filter analysis and and provided the automated non-targeted method for LC-Orbitrap MS analysis. YS conducted the data analysis and wrote the manuscript with contribution from all co-authors.

## Acknowledgements

The Manchester Aerosol Chamber acknowledges the funding support from the European Union's Horizon 2020 research and
innovation programme under grant agreement no. 730997, which supports the EUROCHAMP2020 research programme. Instrumentational support was funded by the NERC Atmospheric Measurement and Observational Facility (AMOF). Y.W. acknowledges the joint scholarship of The University of Manchester and Chinese Scholarship Council. M.R.A. acknowledges funding support by UK National Centre for Atmospheric Sciences (NACS). A.V. acknowledges the funding support by Natural Environment Research Council (NERC) EAO Doctoral Training Partnership.




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
