# Peer review of "Chemical composition of secondary organic aerosol particles formed from mixtures of anthropogenic and biogenic precursors"

_Atmospheric Chemistry and Physics, 2022_

## Author Comment (AC1)

It is interesting to determine the chemical composition and interactions during SOA formation in mixed VOC systems (photooxidation of  $\alpha$ -pinene, isoprene, o-cresol and their binary and ternary mixtures in the presence of NOx and ammonium sulphate seed particles) by using nontargeted LC-Orbitrap MS. The method is innovative. But more detailed information about the methods can be provided.

We kindly thank the reviewer for their time and effort in providing comments for our manuscript. Please see our responses below (shown in blue).

Introduction:

**What are the pros and cons of using non-targeted LC-Orbitrap MS analysis for data interpretation can be addressed?**

We appreciate the opportunity to expand on the benefits and challenges. We will introduce the advantages of using non-targeted LC-Orbitrap MS analysis for data interpretation in section 1.3, from line 101, which has been changed to ".....application. Non-targeted analysis extracts the chemical information of all detected compounds in a sample dataset, providing tentative identification of unknown compounds via library screening, while allowing the rapid chemical characterisation of complex mixtures through the chemical classification of detected compounds in a given sample Place et al. (2021) and Pereira et. al 2021. Mezcua et al. (2011) reported that 210 pesticides were successfully been detected and identified in 78 positive samples of fruit and vegetable samples by using automatic non-targeted screening method in LC-TOF analysis. High-resolution accurate mass spectrometry (HRAM-MS)-based non-targeted screening analysis were applied in chemical characterized of tobacco smoke, and successfully identified a total of known 331 compounds and 50 novel compounds as being present in the sample (Arndt et al., 2019). "

The challenges associated with its use will be introduced in section 1.3, from line 108; the section has been rephrased to ".....low concentration species. However, non-targeted screening methods are not infallible and rigorous testing of autonomous platforms must be performed to understand potential limitations of these tools. Moreover, it is challenging to make semiquantitative or quantitative measurements of unknown compounds in complex matrices. It is worth to noting that quantitative measurements of unknow compounds is a general limitations of ESI operation and not directly attributed to non-targetd screen method, but arguably become more important. It is difficult to perform quantitative measurement of unknow compounds due to the analytical standards for SOA products are limited and only a few molecules out of the thousands detected compounds might be known. Therefore, it is also challenge to determine sample extraction recoveries during sample extraction procedures. The approach of using the normalized abundance of compounds in the sample does not consider different compound ESI efficiencies, which can be influenced by the molecular structure among other parameters (Priego-Capote and Luque De Castro, 2004). For example, Cech and Enke (2000) found out that ESI response increased for peptides with more extensive non polar region. Cech and Enke (2001) further examined and concluded that analytes with more polar portion has lower ESI response than the more nonpolar analytes. Differences in ESI efficiencies of *individual compounds may impact normalized abundance of chemical groupings, particularly when comparing sample compositions which differ appreciably.*

**Method:**

**There are lots of anthropogenic VOC precursors, why o-cresol was chosen as an anthropogenic precursor in this study?**

The work in this paper is a subset of a more comprehensive chamber study of SOA formation from mixed precursors and here focuses on the chemical composition of the SOA formed using LC-Orbitrap MS. A comprehensive and detailed description about the experimental design of the project is presented in Voliotis et al. (2022), which describes the choice of VOC precursors, and their "representativeness" in section 2.1.

We choose *o*-cresol as a moderate SOA yield anthropogenic precursor with comparable reactivity towards the available oxidants (OH radicals) as the two biogenic VOC in a mixtures (a-pinene and isoprene), such that they each may contribute comparably to the distribution of oxidation products.

Humidity and temperature are important factors for SOA formation, they are controlled by the humidifier and by controlling the air conditioning during the experiment. These parameters should be added in the manuscript.

We included this in section 2.3:

From line 259-261: "Photochemistry was initiated by irradiating the VOC at a moderate VOC / NOx ratio using the lamps as described above. The temperature and relative humidity conditions were controlled at 50 %  $\pm$  5 % and 24  $\pm$  2°C, respectively during the experiment. The concentration of NOx and O3, particles number concentration and mass concentration were monitored during the experiment using the online instruments (Pereira et al., 2021)."

**Why was the mass concentration of seed particle doubled in single isoprene experiment?**

The seed particles were inadvertently added into the chamber with increased mass concentration for the single isoprene system. Whilst this could have resulted in a greater partitioning of oxidation products leading to more SOA particle mass forming, the particle mass and resulting yield in the single precursor isoprene system was negligible (SOA particle mass concentration  $\sim 0$  ug/m3), consistent with other studies using neutral seeds.

**How many repeated experiments performed in each experiment type?**

Three replicate experiments were conducted for all systems except the single precursor isoprene systems. This information has been added in the section 2.4.2, Line 315: "*To provide confidence in the components in each system detected by the non-targeted method, only those*

compounds found in all three replicate experiments (two in the single precursor isoprene and binary o-cresol/isoprene systems) and not found in any background "clean" experiments were attributed to a particular single precursor or mixed system."

**Before filter sampling, any denuder was used to remove VOCs, NOx and oxidants?**

No denuder was used to remove VOCs, NOx and oxidants before filter sampling owing to the challenge associated with gaseous denuding at the high sampling flow rate. Chamber air was flushed out at around 3 m3 min-1 onto the filter, taking some 5 minutes for sample collection. Du et al. (2021) had combined the online (FIGAREO-CIMS) and offline mass spectrometric (LC-Orbitrap MS) techniques to characterize the chemical composition of secondary organic aerosol (SOA) generated from the photooxidation of  $\alpha$ -pinene in the MAC. The study of Du et al. (2021) reported that the distribution of particle-phase products is highly consistent between the FIGAERO-CIMS and LC-Orbitrap MS negative ionisation mode for the  $\alpha$ -pinene SOA products, suggesting near negligible (or at least comparable) gas phase absorption artefact introduced during filter collection in both techniques.

Results and discussion:

Online data from gas chromatography mass spectrometer (GCMS), condensation particles counter, differential mobility particle sizer (DMPS) and aerosol mass spectrometer (AMS) are very useful for data interpretation. But the results were not reported in this study.

As mentioned above, this is part of a more comprehensive study of SOA formation in mixtures. The full instrument description is given in in Voliotis et al. (2022), and the DMPS, GCMS and AMS, along with the online FIGAREO-CIMS, data are presented therein, and in several companion papers (Du et al., 2021; Shao et al., 2022)

The online GCMS data show the decay of precursors at each system in Figure 1(d)-(f). We could not extract more information about the chemical composition of gas-phase products from the online GCMS in our experiments, but the products are reported in Du et al., (2022a,b). The AMS data were utilised to show the evolution of SOA mass of each precursor system and presented in Figure 1(a), but high resolution data are compared in detail with FIGAREO-CIMS data and offline LC-Orbitrap MS data in Shao et al. (2022, in prep).

Lots of data were presented in this study, (e.g. number of detected SOA compounds, molecular composition, compositional analysis). The novel part of this study is about the unique-to-mixture products due to the interactions between VOC products. This section can be extended and provide more mechanistic understanding of their formation.

We thank to the reviewer for this suggestion. To provide more mechanistic understanding of their formation required structure identification and quantification of the unique-to-mixture compounds, which require standards. Mechanistic inferences are provided in the combined use of FIGAREO-CIMS data and offline LC-Orbitrap MS in Du et al. (2022a,b) and further work to elaborate on the potential mechanisms is recommended in line with these studies.

We include this at the end Conclusion section: "This study did not examine the molecular structure of the unique compounds/potential tracers in the mixture precursors systems. The

future studies suggest focus on identifying the molecular structure of unique-to-mixture components will help better understand the detailed mechanisms of interactions involved in ambient SOA formation from mixture VOC oxidations."

**Reference:**

Arndt, D., Wachsmuth, C., Buchholz, C., and Bentley, M.: A complex matrix characterization approach, applied to cigarette smoke, that integrates multiple analytical methods and compound identification strategies for non-targeted liquid chromatography with high-resolution mass spectrometry, Rapid Communications in Mass Spectrometry, 34, 10.1002/rcm.8571, 2019.

Cech, N. and Enke, C.: Practical Implications of Some Recent Studies in Electrospray Ionization Fundamentals, Mass spectrometry reviews, 20, 362-387, 10.1002/mas.10008, 2001.

Cech, N. B. and Enke, C. G.: Relating Electrospray Ionization Response to Nonpolar Character of Small Peptides, Analytical Chemistry, 72, 2717-2723, 10.1021/ac9914869, 2000.

Du, M., Voliotis, A., Shao, Y., Wang, Y., Bannan, T. J., Pereira, K. L., Hamilton, J. F., Percival, C. J., Alfarra, M. R., and McFiggans, G.: Combined application of Online FIGAERO-CIMS and Offline LC-Orbitrap MS to Characterize the Chemical Composition of SOA in Smog Chamber Studies, Atmos. Meas. Tech. Discuss., 2021, 1-42, 10.5194/amt-2021-420, 2021. Mezcua, M., Malato, O., Martinez-Uroz, M. A., Lozano, A., Agüera, A., and Fernández-Alba, A. R.: Evaluation of Relevant Time-of-Flight-MS Parameters Used in HPLC/MS Full-Scan

Screening Methods for Pesticide Residues, Journal of AOAC INTERNATIONAL, 94, 1674-1684, 10.5740/jaoacint.SGEMezcua, 2011.

Pereira, K. L., Ward, M. W., Wilkinson, J. L., Sallach, J. B., Bryant, D. J., Dixon, W. J., Hamilton, J. F., and Lewis, A. C.: An Automated Methodology for Non-targeted Compositional Analysis of Small Molecules in High Complexity Environmental Matrices Using Coupled Ultra Performance Liquid Chromatography Orbitrap Mass Spectrometry, Environmental Science & Technology, 55, 7365-7375, 10.1021/acs.est.0c08208, 2021.

Place, B. J., Ulrich, E. M., Challis, J. K., Chao, A., Du, B., Favela, K., Feng, Y.-L., Fisher, C. M., Gardinali, P., Hood, A., Knolhoff, A. M., McEachran, A. D., Nason, S. L., Newton, S. R., Ng, B., Nuñez, J., Peter, K. T., Phillips, A. L., Quinete, N., Renslow, R., Sobus, J. R., Sussman, E. M., Warth, B., Wickramasekara, S., and Williams, A. J.: An Introduction to the Benchmarking and Publications for Non-Targeted Analysis Working Group, Analytical Chemistry, 93, 16289-16296, 10.1021/acs.analchem.1c02660, 2021.

Priego-Capote, F. and Luque de Castro, M. D.: Analytical uses of ultrasound I. Sample preparation, TrAC Trends in Analytical Chemistry, 23, 644-653,

https://doi.org/10.1016/j.trac.2004.06.006, 2004.

Shao, Y., Voliotis, A., Du, M., Wang, Y., Pereira, K., Hamilton, J., Alfarra, M. R., and McFiggans, G.: Chemical composition of secondary organic aerosol particles formed from mixtures of anthropogenic and biogenic precursors, Atmos. Chem. Phys. Discuss., 2022, 1-41, 10.5194/acp-2022-127, 2022.

Voliotis, A., Du, M., Wang, Y., Shao, Y., Alfarra, M. R., Bannan, T. J., Hu, D., Pereira, K. L., Hamilton, J. F., Hallquist, M., Mentel, T. F., and McFiggans, G.: Chamber investigation of the formation and transformation of secondary organic aerosol in mixtures of biogenic and anthropogenic volatile organic compounds, Atmos. Chem. Phys. Discuss., 2022, 1-49, 10.5194/acp-2021-1080, 2022.

---

## Author Comment (AC2)

**General Comments:**

This work by Shao et al. is a follow up to the work by McFiggans et al. in Nature 2019 on the impacts of mixed VOC systems on SOA formation. They performed a series of batch mode chamber experiments with single and mixed precursors of biogenic and anthropogenic origin in the presence of $NO_x$ and aerosol seed. Offline analysis of the SOA composition was performed primarily with LC-MS to elucidate which species dominate and dictate the SOA formation in mixtures and identify any cross products. This work is novel and of value to the community, although I find it to be overly verbose and rambling and suggest editing to make it more concise and flow better if possible. This is appropriate for ACP after addressing the other suggestions below.

**Specific Comments:**

Line 134: I think this is well established and suggest re-wording "might be the reason" to something more definitive

Line 134 revised from "which *might be the reason*" to "*which is the reason…*".

Line 207: What is the residence time in the chamber?

The MAC is operated as a batch reactor and is not operated in "flow through" mode. The air is held continuously after introduction, as described in Shao et al. (2022), so essentially has an infinite residence time (though component lifetimes clearly would be limited by the losses to, and interactions with, the Teflon walls). The experimental duration is typically 6 hours, though experiments up to 24 hours are possible.

Section 2.4.1: Is it possible for chemical transformations to occur during the 2 hr ambient temperature rest, sonication, or drying? Would this be observable? Can you comment on how this may impact results?

It is difficult to determine the quality of the aerosol extraction procedure using non-targeted analysis due to the difficulty of unknown compound identification. The analytical standards for SOA products are limited and only a few molecules out of the thousands detected compounds might be known. Therefore, it is also difficult to determine sample extraction recoveries during sample extraction procedures, since various compounds will have different recovery efficiencies that can be influenced by the molecular structure (Priego-Capote and Luque De Castro, 2004).However, we note that the sample extraction procedure performed in this study is common practice for the analysis of OA, with the majority of studies using either water or methanol as the extraction solvent, followed by sonication and evaporation(Gao et al., 2006; Hamilton et al., 2008; Kourtchev et al., 2016)  It should be noted that we used methanol as extraction solvent since sonication using water can result in the formation of hydroxyl radicals (Miljevic et al., 2014). Moreover, a procedure-controlled sample using a blank filter subject to the same extraction procedure, was performed and analysed using LC-Orbitrap MS analysis. Any artefacts introduced into the samples during sample preparation were excluded from the sample data (see Pereira et. a. 2022 for further information)..  However, This does not provide insights into potential chemical transformations in the OA samples. Investigating

potential chemical transformations during the preparation of the OA samples using non-targeted screening would be incredibly challenging. A control sample (*i.e.* a portion of the same OA sample to allow comparison) would still need to undergo some form of extraction into solvent to allow LC-MS analysis. Further, evaporation and resuspension of the sample into a smaller volume (i.e. sample concentration) will almost certainly be required to allow detection of the trace-level compounds present in OA. However, individual compounds within OA could be targeted to investigate possible chemical transformations, spiking a known quantity of a chemically labelled authentic standard (e.g. deuterated) into the OA sample before solvent extraction. This would allow any chemical transformations to be observed and the extraction recovery of the spiked compound to be determined. We recommend in Pereira et. al. 2022 investigating the recovery efficiencies of authentically identified compounds in future work to quantify any potential losses and provide insights into the quality of the extraction procedure. The work presented here was performed prior to the publication of Pereira et. al. 2022 and subsequently does not include this in investigation.

Line 304: Where would the sodium and potassium come from?

Sodium and potassium could come from several sources during sampling preparation and analysis, such as the mobile phase additives, solvent impurities and so on. The main source is leaching from glassware used to prepare the solvents (Kruve and Kaupmees, 2017). The sentence in the line 302- 304 rephrased to

"*The method provides molecular formulae assignment of detected compounds using the following elemental restrictions: unlimited carbon, hydrogen and oxygen atoms, up to 5 nitrogen and sulphur atoms, and in positive ionisation mode, 2 sodium and 1 potassium atom are also allowed (sodium and potassium are typically introduced into the samples via glassware).*"

Figure 2: Are these common molecular *structures* or molecular *composition*?

These are common discrete molecules. We compared the molecule list between different precursor systems and identified the common molecules which were only considered to be the same detected molecular species if they had a retention time within 0.1 minutes.

The captions in Figure 2 rephased to " *Number of common discrete molecules and unique compounds in single and binary precursors mixed experiments detected by negative ionization mode LC-Orbitrap MS. Product are considered identical in the mixed and single precursor systems if the molecules has the same empirical formula and a retention time difference <0.1min.*"

Also, the caption in Figure 3 rephased to "*Number of common discrete molecules and unique compounds in single and binary precursors mixed experiments detected by positive ionization mode LC-Orbitrap MS. Product are considered identical in the mixed and single precursor systems if the molecules has the same empirical formula and a retention time difference <0.1min.*"

Figure 4: Why does essentially all the signal contain nitrogen for cresol and any mixtures with cresol? This is discussed ~line 520 but not the reasoning for why N-containing species are highly dominant.

Nitrogen containing compounds, such as those containing amide groups or nitro groups, effiencently ionize using electrospray ionization (Oss et al., 2010). As the MCM v3.3.1 and our study shown, *o*-cresol photo oxidation produces a number of different kinds of nitro-aromatic compounds, such as methyl-nitrocatechol and methyl-nitrophenol. These compounds have high negative mode sensitivity using electrospray ionisation, contributing to substantial signal in the systems that contain *o*-cresol.

This is mentioned in line 526-528, line 576-577, line 783-784, line 851-852, and line 954-956 in the manuscript. Reference ( https://doi.org/10.1021/ac902856t ) will add in line 528 to further support the explanation.

Line 467: Suggest using HOM definition from Bianchi et al (https://pubs.acs.org/doi/10.1021/acs.chemrev.8b00395): highly oxygenated organic molecules

The definition of HOM in line 467 had been rephased to "highly *oxygenated organic molecules*"

Lines 466-472: This section on HOM is not well fleshed out and doesn't seem to flow with the discussion. Suggest removing or re-writing. Please add a reference for this sentence, or remove: "Autoxidation may therefore contribute to CHO products with carbon numbers 16 – 20 in α-pinene oxidation"

We rephased the sentences from line 466 to 472 as shown below:

"*Autoxidation of RO₂ radicals in the gas-phase occurs rapidly via inter/intramolecular hydrogen abstraction leading to forming R radicals with subsequent O₂ addition (Mentel et al., 2015; Jokinen et al., 2014). The new RO₂ radicals can undergo further autoxidation reaction, or react with RO₂ to generate dimer accretion products (Zhao et al., 2018; Berndt et al., 2018), leading to so-called highly oxygenated organic molecules (HOM) with very-low volatilities (Bianchi et al., 2019). Autoxidation may therefore contribute to CHO products with carbon numbers 16 – 20 in α-pinene oxidation (Berndt, 2021; Ehn et al., 2014)*"

Line 477: Please state how much SOA was formed. It is confusing that this line (and above) states ~0 μg/m3 was formed but the section goes on to discuss the compounds measured in the particle phase

The SOA particle mass was 0.1 μg/m³ in single precursor isoprene system. The sentence in line 477 rephrased to "*As also seen in Fig.1(a), negligible SOA particle mass was generated in the single precursor isoprene system (0.1g/m³, close to our chamber background*". We measured the SOA particle mass by using the online HR-TOF-AMS, which detected negligible mass in the chamber, at the same order as the background chamber levels. This does not mean that there were no particulate components derived from isoprene present, just that the total mass was practically indistinguishable from the background mass using our online instrumentation. The compound measurements we report were obtained from offline LC-orbitrap MS analysis, with its ability to detect compounds with trace sensitivity much lower than the limit of detection of the AMS. The compounds could be detected by LC-Orbitrap MS, and use automate non –

targeted screening method to assign the molecular formulae as long as the masses error < 3 ppm, signal-to-noise ratio > 3, and the isotopic intensity tolerance was within ± 30 % of the measured and theoretical isotopic abundance.

Lines 493-494: This doesn't reflect the current state of knowledge and is an insufficient explanation/discussion. Several recent studies have shown that small particles are detected in SOA as a result of decomposition, typically via thermal processes, during analysis. While this work doesn't utilize heating techniques, it does involve substantial sample prep (see comment on section 2.4.1).

As stated in the response for the comment on section 2.4.1, it is difficult to determine the likelihood of potential chemical transformation in the OA sample during the aerosol extraction procedure. An extra sentence will be added at line 495:

"*The possibility that small detected molecules were formed in the filter sample extraction process cannot be ruled out. For example, degradation of organic compounds can be induced by ultrasonic extraction of particulate matter from filters (Miljevic et al., 2014; Mutzel et al., 2013),*"

Lines 496-498: I'm confused why the experiment would be designed in a way that is well documented to not make SOA when the stated point of this work is to make SOA and measure the particle phase composition? Please explain the reasoning for this experimental design and how this advances our understanding of multi-component SOA formation.

Our experimental program aims to establish a framework to understand interactions in systems of mixed anthropogenic and biogenic VOCs. Therefore, we choose precursors considering the potential diversity in VOC sources contributing to the ambient atmosphere, building on the previous insight from McFiggans et al.(2019) which used a binary mixture of biogenic low yield (isoprene) and high yield (α-pinene) precursors. The aim is to further investigate the interaction in systems of mixed low, moderate or high yield VOCs, with both anthropogenic or biogenic species able to compete for the available OH.

There is a clear indication of suppression of the yield of α-pinene in its mixture with isoprene, but as with the *o*-cresol / isoprene mixture, there is a possible indication of enhancement, though this is too small to be unambiguous. In the ternary system, it is unclear if there is a suppression or enhancement effect with regard to measured total SOA particle mass, but chemical interactions are evident from the unique-to mixture components. All these results about SOA yield across different precursors systems, details of experimental design, and the complexity of the systems introduced substantial challenges to their interpretation were comprehensively elucidated in the companion paper Voliotis et al. (2022).

We still could measure and analyse the SOA particulate product from isoprene, which is well documented to make little SOA particle mass under neutral seed conditions, by use of the LC-Orbitrap MS technique and its unprecedented high-resolution accurate mass (HRAM) allowing unambiguous identification of molecular formulae. .

Line 501: Can you be sure these species are created from isoprene + OH and not impurities in your isoprene source or chamber contamination?

The isoprene gas precursor was introduced into the chamber by injecting liquid isoprene (Sigma-Aldrich, purity >=99%) through a rubber seal into a glass bulb that temperature was kept at approximately 100°C, and flush to the chamber by using nitrogen as carrier gas. The glass bulb was warm and flushed with $N_2$ during the "pre-experiment" protocol before the conducted experiment ensuring contamination in the glass bulb was flushed out and maintain cleanness. Thus, we are confident that these species are not contamination or impurities in the isoprene source.

Our chamber had conducted off-gasing and actinometry experiments regularly to determine the chamber background contamination. Filter samples also been collected after each off-gasing and actinometry experiment, that followed the same sample extraction procedure and LC-Orbitrap MS analysed method. None of the reported compounds were found in these background experiments and any contamination from chamber introduced into the experimental samples can therefore be excluded.

Line 679: Here you mention the possibility of fragmentation of larger species resulting in the smaller species measured in the particle phase. Please include references (e.g. https://pubs.acs.org/doi/abs/10.1021/acs.est.5b04769).

The reference (Lopez-Hilfiker et al., 2016) will be added into the text of manuscript.

Line 765: It isn't clear to me that accretion reactions have occurred during SOA formation rather than alterations during sample prep and analysis. Additionally, if they did occur during the experiment, can you be sure that accretion products would still form under atmospherically relevant precursor and SOA concentrations?

The intensity weighted average values of nC clearly show accretion products to be present in the single precursor isoprene system, the binary α -pinene-containing systems and the ternary system in positive ionization mode. It is indeed difficult to guarantee that the OA chemical transformation did not happen during sample preparation. Therefore, the sentence in line 765 has been rephrased to

"*It is apparent in the positive mode that accretion reactions occurred, and its products play an essential role in the single precursor isoprene system, the binary α -pinene-containing systems and the ternary system. It cannot be discounted that chemical transformation may occur during filter sample preparation, which might impact on the intensity weighted average values of various chemical properties.*"

Some of the accretion products, such as $C_{21}H_{33}NO_4$ in binary α-pinene/o-cresol system, $C_9H_{11}NO$, and $C_8H_8O_{10}$ in binary isoprene/*o*-cresol system were found uniquely in the mixed precursors system with non-negligible normalized signal abundance, enabling their use as tracers in the ambient environment.

**Technical:**

Throughout manuscript: NOx should have a subscript "x" and be $NO_x$

Changed

Throughout manuscript: change instances of "ml" to "mL"

Changed

Throughout manuscript: change instances of "ug" to "µg"

Changed

Line 222: particles counter à particle counter (plural to singular)

Changed

Reference:
Berndt, T.: Peroxy Radical Processes and Product Formation in the OH Radical-Initiated Oxidation of α-Pinene for Near-Atmospheric Conditions, The Journal of Physical Chemistry A, 125, 9151-9160, 10.1021/acs.jpca.1c05576, 2021.
Bianchi, F., Kurtén, T., Riva, M., Mohr, C., Rissanen, M. P., Roldin, P., Berndt, T., Crounse, J. D., Wennberg, P. O., Mentel, T. F., Wildt, J., Junninen, H., Jokinen, T., Kulmala, M., Worsnop, D. R., Thornton, J. A., Donahue, N., Kjaergaard, H. G., and Ehn, M.: Highly Oxygenated Organic Molecules (HOM) from Gas-Phase Autoxidation Involving Peroxy Radicals: A Key Contributor to Atmospheric Aerosol, Chemical Reviews, 119, 3472-3509, 10.1021/acs.chemrev.8b00395, 2019.
Ehn, M., Thornton, J. A., Kleist, E., Sipilä, M., Junninen, H., Pullinen, I., Springer, M., Rubach, F., Tillmann, R., Lee, B., Lopez-Hilfiker, F., Andres, S., Acir, I.-H., Rissanen, M., Jokinen, T., Schobesberger, S., Kangasluoma, J., Kontkanen, J., Nieminen, T., Kurtén, T., Nielsen, L. B., Jørgensen, S., Kjaergaard, H. G., Canagaratna, M., Maso, M. D., Berndt, T., Petäjä, T., Wahner, A., Kerminen, V.-M., Kulmala, M., Worsnop, D. R., Wildt, J., and Mentel, T. F.: A large source of low-volatility secondary organic aerosol, Nature, 506, 476-479, 10.1038/nature13032, 2014.
Gao, S., Surratt, J., Knipping, E., Edgerton, E., Shahgholi, M., and Seinfeld, J.: Characterization of polar organic components in fine aerosols in the southeastern United States: Identity, origin, and evolution, Journal of Geophysical Research, 111, 10.1029/2005JD006601, 2006.
Hamilton, J. F., Lewis, A. C., Carey, T. J., and Wenger, J. C.: Characterization of Polar Compounds and Oligomers in Secondary Organic Aerosol Using Liquid Chromatography Coupled to Mass Spectrometry, Analytical Chemistry, 80, 474-480, 10.1021/ac701852t, 2008.
Kourtchev, I., Godoi, R. H. M., Connors, S., Levine, J. G., Archibald, A. T., Godoi, A. F. L., Paralovo, S. L., Barbosa, C. G. G., Souza, R. A. F., Manzi, A. O., Seco, R., Sjostedt, S., Park, J. H., Guenther, A., Kim, S., Smith, J., Martin, S. T., and Kalberer, M.: Molecular composition of

organic aerosols in central Amazonia: an ultra-high-resolution mass spectrometry study, Atmos. Chem. Phys., 16, 11899-11913, 10.5194/acp-16-11899-2016, 2016.

Kruve, A. and Kaupmees, K.: Adduct Formation in ESI/MS by Mobile Phase Additives, Journal of The American Society for Mass Spectrometry, 28, 887-894, 10.1007/s13361-017-1626-y, 2017.

Lopez-Hilfiker, F. D., Mohr, C., D'Ambro, E. L., Lutz, A., Riedel, T. P., Gaston, C. J., Iyer, S., Zhang, Z., Gold, A., Surratt, J. D., Lee, B. H., Kurten, T., Hu, W. W., Jimenez, J., Hallquist, M., and Thornton, J. A.: Molecular Composition and Volatility of Organic Aerosol in the Southeastern U.S.: Implications for IEPOX Derived SOA, Environmental Science & Technology, 50, 2200-2209, 10.1021/acs.est.5b04769, 2016.

Miljevic, B., Hedayat, F., Stevanovic, S., Fairfull-Smith, K. E., Bottle, S. E., and Ristovski, Z. D.: To Sonicate or Not to Sonicate PM Filters: Reactive Oxygen Species Generation Upon Ultrasonic Irradiation, Aerosol Science and Technology, 48, 1276-1284, 10.1080/02786826.2014.981330, 2014.

Mutzel, A., Rodigast, M., Iinuma, Y., Böge, O., and Herrmann, H.: An improved method for the quantification of SOA bound peroxides, Atmospheric Environment, 67, 365-369, https://doi.org/10.1016/j.atmosenv.2012.11.012, 2013.

Oss, M., Kruve, A., Herodes, K., and Leito, I.: Electrospray Ionization Efficiency Scale of Organic Compounds, Analytical Chemistry, 82, 2865-2872, 10.1021/ac902856t, 2010.

Priego-Capote, F. and Luque de Castro, M. D.: Analytical uses of ultrasound I. Sample preparation, TrAC Trends in Analytical Chemistry, 23, 644-653, https://doi.org/10.1016/j.trac.2004.06.006, 2004.

Voliotis, A., Du, M., Wang, Y., Shao, Y., Alfarra, M. R., Bannan, T. J., Hu, D., Pereira, K. L., Hamilton, J. F., Hallquist, M., Mentel, T. F., and McFiggans, G.: Chamber investigation of the formation and transformation of secondary organic aerosol in mixtures of biogenic and anthropogenic volatile organic compounds, Atmos. Chem. Phys. Discuss., 2022, 1-49, 10.5194/acp-2021-1080, 2022.

---

## Author Response (AR2)

This study determined the photooxidation of single and mixed biogenic (isoprene and  $\alpha$ -pinene) and anthropogenic (o-cresol) precursors in the presence of NOx and seed particles. It is interesting to determine the chemical interactions in mixed VOC systems by using non-targeted LC-Orbitrap MS. This work is novel and provides some interesting data, but the methodology and discussion can be described in detail. There are still some comments need to be addressed:

1) Line 211-213: Isoprene is the most abundant biogenic VOC emission and  $\alpha$ -pinene is one of the most abundant and widely studied biogenic monoterpene. There are lots of anthropogenic VOC precursors, e.g., BTEX, which are the most abundant aromatic VOCs. But why o-cresol was chosen as an anthropogenic precursor was not described in this and previous studies (Voliotis et al, 2022).

The primary reason for selection of *o*-cresol was its comparable reactivity toward the OH radical with those of isoprene and  $\alpha$ -pinene, allowing *o*-cresol to have initial isoreactivity at concentrations comparable to the other precursors so that the oxidation products from each precursor are likely to be of comparable abundance in a mixed VOC system. Other anthropogenic VOC precursors such as benzene, ethylbenzene, o-xylene, p-xylene, m-xylene, and toluene ("BTEX"), whilst more abundant in the ambient atmosphere, have a significantly lower rate coefficient toward OH than the two biogenic VOCs as shown in the table below (Atkinson, 2004; Calvert, 2002). This would require substantially higher concentrations of each, compromising our experimental design.

| VOC          | Rate coefficient at 298K (k/cm 3 molecule -1 s -1 ) |
|--------------|--------------------------------------------------------------------------------------|
|              |                                                                                      |
| Benzene      | $1.2 \text{ x} 10^{-12}$                                                             |
| Ethylbenzene | $7.0 \ge 10^{-12}$                                                                   |
| o-xylene     | 1.36 x 10 -11                                                             |
| p-xylene     | 1.43 x 10 -11                                                             |
| m-xylene     | 2.31 x 10 -11                                                             |
| toluene      | 5.60 x 10 -12                                                             |
| o-cresol     | 4.10x 10 -11                                                              |
| α-pinene     | 5.30 x 10 -11                                                             |
| isoprene     | 1x 10 -10                                                                 |

This choice does not devalue the study. *o*-cresol is an anthropogenic VOC with a moderate SOA yield that is directly emitted to the ambient environment during coal, wood, and municipal solid waste combustion and is an oxidation product of toluene (e.g. from automobile exhaust), which is among the most abundant anthropogenic hydrocarbons in the atmosphere (Deng et al., 2017; Zhao et al., 2004)

This information has been added in lines 211-213: "Ortho-cresol (o-cresol) was chosen as an anthropogenic precursor with a moderate SOA yield, between that of isoprene and  $\alpha$ -pinene. o-cresol has reactivity toward the hydroxyl radical (OH) that is comparable to those of the chosen biogenic VOCs (Atkinson, 2004) and a negligible reactivity towards ozone. Hence, the oxidation products from each precursor are likely to be of comparable abundance in a mixed systems."

2) Line 284-286: As no denuder was used to remove VOCs and oxidants, particle collected on filters will/may perform further oxidation (or reaction). Would this be observable in this study? How to prove that no further reactions have been performed on the filter? Can you comment on how this may impact results?

We thank the reviewer for pointing this out. Filter sampling can exhibit both negative and positive artefacts. Gas-phase organic compounds can be adsorbed on the filter medium and on particle mass collected during sampling, whilst volatilisation of "non-involatile" particulate components can reduce the collected mass. Adsorption leads to the overestimation of organic particle mass, while volatilisation results in its underestimation. It is difficult to isolate and quantify these two artefacts since they can occur simultaneously (Subramanian et al., 2004). Additionally, organic particulate products may be formed by further reaction on the filter (Cheng et al., 2009), changing the chemical composition of collected particulate matter.

Our samples were rapidly collected (emptying the chamber through the filter in 5 or 6 minutes) and it is not possible to effectively denude the gases at the flow rate through the filter in our chamber. Whilst this leads to limited time for adsorption and reaction, it is difficult to quantify these impacts. It should be noted that the formation of products from such reactions could also occur as a result of gas-particle collisions during the experiment during the much longer residence time in the chamber. A companion paper in this project characterising the chemical composition using online (FIGAREO-CIMS) and offline mass spectrometric (LC-Orbitrap MS) techniques showed high consistency between the online I-CIMS and offline negative ionization mode (Du et al., 2021) within the limitations of the techniques. Whilst both techniques use filter collection, this is indicative that the further reactions on the filter after collection at very different flowrates and sampling times are comparable (or negligible). Further work is required to fully quantify any such artefacts.

A short discussion to this end has been added in line 294 in manuscript: " It is noted that *both positive (conversion of gas phase organics to particulate form) and negative (volatilisation of particulate organic compounds) artefacts are possible during collection of particulate matter during filter sampling, resulting in overestimation and underestimation of particulate organic carbon, respectively. The samples were rapidly collected in our experiments (emptying the chamber through the filter in 5 or 6 minutes), precluding the ability to effectively denude gases at the flow rate. Whilst gases may be adsorbed / adsorbed on the filters, it is challenging to quantify these impacts. Formation of products of reactions in the particles themselves could also occur after to gas-particle collisions during the experiment with a much longer residence time in the chamber.*

Du et al. (2021) combined the online (FIGAREO-CIMS) and offline mass spectrometric (LC-Orbitrap MS) techniques to characterise the chemical composition in the same systems. It was reported that the distribution of particle-phase products is highly consistent between the I-CIMS and LC-Orbitrap MS negative ionisation mode for the  $\alpha$ -pinene SOA products, suggesting near negligible (or at least comparable) gas phase absorption artefacts introduced during filter collection in both techniques."

3) Line 298-304: Any recovery test has been performed? If yes, please provide more information about the extraction recoveries.

No recovery test was performed in this study.

This information will be added in the line 324: "The efficiency of the aerosol extraction procedure using non-targeted analysis in this study is difficult to determine owing to the limitation of unknown compound identification. Few molecules of the thousands detected can be identified in the analytical standards for SOA products. It is also difficult to determine sample extraction recoveries since compounds have different recovery efficiencies determined by their molecular structure (Priego-Capote and Luque De Castro, 2004). Much further work on the recovery efficiency is required to quantify potential losses and provide insights into the quality of the extraction procedure."

4) Line 360-361: "The SOA particle mass continued to increase at the end of the experiment in the single VOC o-cresol and binary isoprene/o-cresol systems". Have the authors tried to obtain the maximum value in the above systems?

Our experiments were not sufficiently long to capture the maximum SOA particle mass in the single VOC *o*-cresol and binary isoprene / *o*-cresol systems because of logistical constraints.

5) Figure1b&c: Only 6 lines were observed for the 7 experiments.

We thank the reviewer for pointing out our mistake and we have replaced figure 1 with the new one, below.

Figure 1: Evolution of gas and total SOA particle mass measurements during the photo-oxidation of VOCs after chamber illumination. (a) The SOA mass was measured using a high-resolution time-of-flight aerosol mass spectrometer (HR-ToF-AMS) during single, binary and ternary experiment. (b)–(c): Concentration of NOx and O3 against time in all of single, binary and ternary experiments. (d)-(f): decay rata of VOC across all systems in.  $\alpha$ -pinene (b), isoprene(c) and o-cresol(d) in single, binary and ternary experiments respectively

6) Line 665-668: There is a generally a greater fraction of the positive ionisation mode signal that is inconclusive than in negative ionisation mode. But the explanations are not provided in details.

"Inconclusive" compounds refer to compounds that were detected in only one of the replicate experiments performed for each precursor system in this study. There are several reasons why a compound may not be detected in both replicate experiments. For example, any compounds with a measured peak area close to the limit of detection (defined as  $3 \times$  signal-to-noise ratio), a minimum peak intensity below  $3 \times 10^4$  and that were not detected in three consecutive scans, were removed from the sample data, and subsequently reported as not-detected. These parameters are used for chromatographic peak detection, minimising the detection/inclusion of erroneous background signal. Second, each molecular formula assignment is based on isotopic pattern and abundance scoring, where the theoretical and measured isotopic pattern and abundance are evaluated. Any compound with an isotopic abundance outside of  $\pm 30\%$  with a mass tolerance greater than 3 ppm will be removed from the sample data. Further, any compound detected in the sample and chamber background experiment with the same molecular formula and retention time  $(\pm 0.1)$  minutes and a sample/chamber background peak area ratio <3 were removed from the sample data. Thus, detected compounds with a measured peak area close to these cut-off values may only be detected in one replicate experiment due to instrumental/experimental variation.

It is worth noting that positive ionisation mode may also be affected by the molecular adduct detected. The non-targeted method cannot detect a sodiated molecular species, if the protonated adduct is not detected (see Pereira et. al (2022) for further information). The protonated molecular species may only represent a small proportion of the measured compound signal and if close to the cut-off values described above, the protonated adduct may not be detected, resulting in the compound being removed from the sample data. This may explain why a greater number of inconclusive compounds were observed in positive rather than negative ionisation mode.

The detailed explanation will be added in line 698: " .... or a larger fraction of the signal from compounds also found on chamber background filters. Moreover, the greater fraction of "inconclusive" compounds in positive ionization mode might also attributed to automated non-targeted method programming. For example, the automated non-targeted method programmed that compound will be removed from the final detected molecules peak list when they have a signal-to-noise ratio below 3 and low measured signal abundance close to the signal-to-noise cut-off values in the replicate experiment. The automated non-targeted method also programmed the molecular formula assignment base on the isotopic pattern, where the isotopic intensity tolerance was within  $\pm 30\%$  of the theoretical isotopic abundance. Consequently, it becomes a challenge to accurately assign a molecular formula to compounds with "large" molecular weights due to arising the number of possible formulars. The "large" compound could have different molecular formula assignments in "representative" and replicate experiments, respectively, though it has a similar retention time and molecular weight in both experiments."

7) Results and Discussion: In this study, the molecular structure of the unique compounds or potential tracers were not determined. We know that are some limitations of using nontargeted LC-Orbitrap MS but there are lots of data from gas chromatography mass spectrometer (GCMS), condensation particles counter, differential mobility particle sizer (DMPS) and aerosol mass spectrometer (AMS) (or further detail chemical analysis) which can help to better interpret the results. Even, there is a clear indication of suppression of the yield of  $\alpha$ -pinene in its mixture with isoprene in this study, but the explanation/discussion were not sufficient, e.g. why nitrogen-containing species are highly dominant in single cresol oxidation? why greater fraction of the positive ionisation mode signal that is inconclusive than in negative ionisation mode?

This study is part of a more comprehensive study of SOA formation in mixtures, and the scope of this manuscript is to understand the chemical composition and interactions during SOA formation in mixed VOC systems by using offline LC-Orbitrap-MS. Comprehensive information about the suppression of the yield of  $\alpha$ -pinene in its mixture with isoprene is presented in Voliotis et al. (2022b), along with the DMPS, GCMS, AMS and FIGAREO-CIMS data in the mixture presented therein. These data are also presented in several companion papers in order to investigate the volatility distribution of products in the mixed system (Voliotis et al., 2021) and explore the chemical compositions in the mixed experiments by the combination of online FIGAERO-CIMS and offline LC-Orbitrap MS measurements (Du et al., 2022). A further study in preparation probes the average carbon oxidation state of SOA in a mixed precursor system by using three different mass spectrometry techniques (AMS, FIAGREO-CIMS and LC-Orbitrap-MS) in order to provide comprehensive insight into the overall effect about molecular interaction impacts the SOA formation and properties. Further explanations about why greater fraction of the positive ionisation mode signal that is inconclusive than in negative ionisation mode will be added in line 673 as mentioned in comment 6 above.

A new section named "3.2.4 Further insight from companion papers" was added in line 1008.

"This study probes the chemical composition and interactions during SOA formation in mixed VOC systems using the offline LC-Orbitrap-MS technique. The complete instrument description and experimental design is given in Voliotis et al. (2022b), along with the data from online techniques (e.g. SMPS, Semi-continuous GCMS, HR-ToF-AMS and FIGAREO-CIMS). Comprehensive analysis of FIGAREO-CIMS and HR-ToF-AMS data is provided in (Voliotis et al., 2022a; Voliotis et al., 2021; Du et al., 2022). Voliotis et al. (2022a) and Voliotis et al. (2021) investigated the volatility distribution of products in mixed systems using the FIGAERO-CIMS and a thermal denuder coupled with an SMPS and HR-ToF-AMS. Voliotis et al. (2021) reported FIGAERO-CIMS measurements showing an abundance of products uniquely found in the  $\alpha$ -pinene / o-cresol mixture, with the majority in the nC=5-10 and nC>10 classes. This result is consistent with the finding in this study that unique compounds were found in  $\alpha$ -pinene / o-cresol mixture obtained from LC-Orbitrap-MS measurement, likely the cross-products from  $\alpha$ -pinene and o-cresol oxidation in the particle-phase. Voliotis et al. (2021) observed a dominant contribution of nitrogen-containing compounds to the total signal in all o-cresol containing systems, similar to the results obtained from negative ionization mode in *LC-Orbitrap-MS in this study. This is unsurprising owing to the high sensitivity of the iodide* CIMS towards o-cresol photo-oxidation produced nitro-aromatic compounds with hydroxyl

groups, such as methyl-nitrocatechol and methyl-nitrophenol (Lee et al., 2014; Iyer et al., 2016)."

Reference:

Atkinson, R. B., D. L. Cox, R. A.Crowley, J. N.Hampson, R. F.Hynes, R. G.Jenkin, M. E.Rossi, M. J.Troe, J.: Evaluated kinetic and photochemical data for atmospheric chemistry: Volume I - gas phase reactions of Ox, HOx, NOxand SOxspecies, Atmos. Chem. Phys., 4, 1738, 10.5194/acp-4-1461-2004, 2004.

Calvert, J. G.: The mechanisms of atmospheric oxidation of aromatic hydrocarbons, Oxford: Oxford University Press, 2002.

Cheng, Y., He, K. B., Duan, F. K., Zheng, M., Ma, Y. L., and Tan, J. H.: Measurement of semivolatile carbonaceous aerosols and its implications: A review, Environment International, 35, 674-681, https://doi.org/10.1016/j.envint.2008.11.007, 2009.

Deng, W., Liu, T., Zhang, Y., Situ, S., Hu, Q., He, Q., Zhang, Z., Lü, S., Bi, X., Wang, X., Boreave, A., George, C., Ding, X., and Wang, X.: Secondary organic aerosol formation from photooxidation of toluene with NOx and SO2: Chamber simulation with purified air versus urban ambient air as matrix, Atmospheric Environment, 150, 67-76, https://doi.org/10.1016/j.atmosenv.2016.11.047, 2017.

Du, M., Voliotis, A., Shao, Y., Wang, Y., Bannan, T. J., Pereira, K. L., Hamilton, J. F., Percival, C. J., Alfarra, M. R., and McFiggans, G.: Combined application of Online FIGAERO-CIMS and Offline LC-Orbitrap MS to Characterize the Chemical Composition of SOA in Smog Chamber Studies, Atmos. Meas. Tech. Discuss., 2021, 1-42, 10.5194/amt-2021-420, 2021.

Iyer, S., Lopez-Hilfiker, F., Lee, B. H., Thornton, J. A., and Kurtén, T.: Modeling the Detection of Organic and Inorganic Compounds Using Iodide-Based Chemical Ionization, The Journal of Physical Chemistry A, 120, 576-587, 10.1021/acs.jpca.5b09837, 2016.

Lee, B. H., Lopez-Hilfiker, F. D., Mohr, C., Kurtén, T., Worsnop, D. R., and Thornton, J. A.: An Iodide-Adduct High-Resolution Time-of-Flight Chemical-Ionization Mass Spectrometer: Application to Atmospheric Inorganic and Organic Compounds, Environmental Science & Technology, 48, 6309-6317, 10.1021/es500362a, 2014.

Priego-Capote, F. and Luque de Castro, M. D.: Analytical uses of ultrasound I. Sample preparation, TrAC Trends in Analytical Chemistry, 23, 644-653, https://doi.org/10.1016/j.trac.2004.06.006, 2004.

Subramanian, R., Khlystov, A. Y., Cabada, J. C., and Robinson, A. L.: Positive and Negative Artifacts in Particulate Organic Carbon Measurements with Denuded and Undenuded Sampler Configurations Special Issue of Aerosol Science and Technology on Findings from the Fine Particulate Matter Supersites Program, Aerosol Science and Technology, 38, 27-48, 10.1080/02786820390229354, 2004.

Voliotis, A., Wang, Y., Shao, Y., Du, M., Bannan, T. J., Percival, C. J., Pandis, S. N., Alfarra, M. R., and McFiggans, G.: Exploring the composition and volatility of secondary organic aerosols in mixed anthropogenic and biogenic precursor systems, Atmos. Chem. Phys. Discuss., 2021, 1-39, 10.5194/acp-2021-215, 2021.

Voliotis, A., Du, M., Wang, Y., Shao, Y., Bannan, T. J., Flynn, M., Pandis, S. N., Percival, C. J., Alfarra, M. R., and McFiggans, G.: The influence of the addition of a reactive low SOA yield

VOC on the volatility of particles formed from photo-oxidation of anthropogenic – biogenic mixtures, Atmos. Chem. Phys. Discuss., 2022, 1-30, 10.5194/acp-2022-312, 2022a.

Voliotis, A., Du, M., Wang, Y., Shao, Y., Alfarra, M. R., Bannan, T. J., Hu, D., Pereira, K. L., Hamilton, J. F., Hallquist, M., Mentel, T. F., and McFiggans, G.: Chamber investigation of the formation and transformation of secondary organic aerosol in mixtures of biogenic and anthropogenic volatile organic compounds, Atmos. Chem. Phys. Discuss., 2022, 1-49, 10.5194/acp-2021-1080, 2022b.

Zhao, L., Wang, X., He, Q., Wang, H., Sheng, G., Chan, L. Y., Fu, J., and Blake, D. R.: Exposure to hazardous volatile organic compounds, PM10 and CO while walking along streets in urban Guangzhou, China, Atmospheric Environment, 38, 6177-6184, https://doi.org/10.1016/j.atmosenv.2004.07.025, 2004.